# Grounding DINO: Marrying DINO with Grounded Pre-Training for Open-Set Object Detection

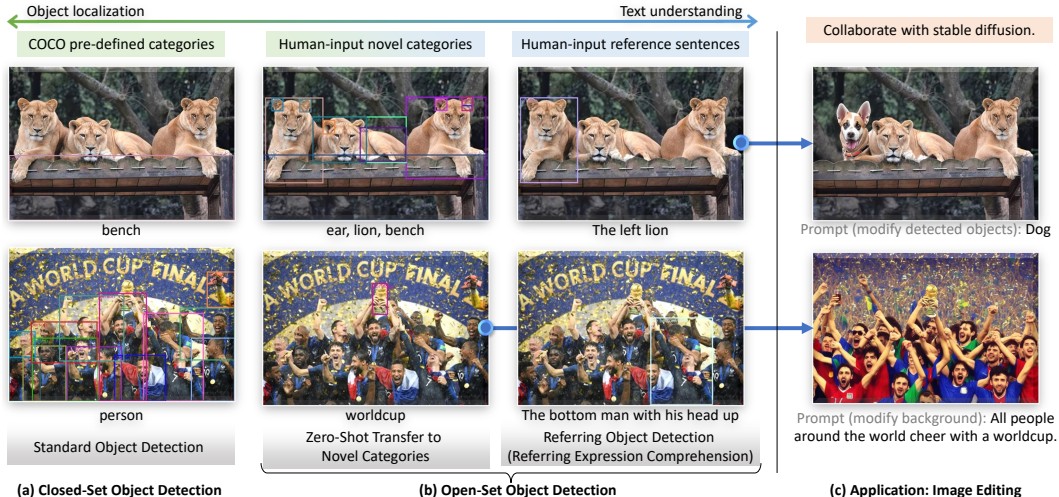

Figure 1: (a) Closed-set object detection requires models to detect objects of pre-defined categories. (b) We evaluate models on novel objects and standard Referring expression comprehension (REC) benchmarks for model generalizations on novel objects with attributes. (c) We present an image editing application by combining Grounding DINO and Stable Diffusion Rombach et al. (2021). Best viewed in colors.

## ABSTRACT

In this paper, we develop an open-set object detector, called Grounding DINO, by marrying Transformer-based detector DINO with grounded pre-training, which can detect arbitrary objects with human inputs such as category names or referring expressions. The key solution of open-set object detection is introducing language to a closed-set detector for open-set concept generalization. To effectively fuse language and vision modalities, we conceptually divide a closed-set detector into three phases and propose a tight fusion solution, which includes a feature enhancer, a language-guided query selection, and a cross-modality decoder for modalities fusion. While previous works mainly evaluate open-set object detection on novel categories, we propose to also perform evaluations on referring expression comprehension for objects specified with attributes. Grounding DINO performs remarkably well on all three settings, including benchmarks on COCO, LVIS, ODinW, and RefCOCO/+/g. Grounding DINO achieves a 52.5 AP on the COCO detection zero-shot transfer benchmark, i.e., without any training data from COCO. It sets a new record on the ODinW zero-shot benchmark with a mean 26.1 AP.

## 1 INTRODUCTION

Understanding novel concepts is a fundamental capability of visual intelligence. In this work, we aim to develop a strong system to detect arbitrary objects specified by human language inputs, which we

name as *open-set object detection*[1]. The task has wide applications for its great potential as a generic object detector. For example, we can cooperate it with generative models for image editing (as shown in Fig. 1 (b)).

The key to open-set detection is introducing language for unseen object generalization (Li et al., 2021; Anderson et al., 2017; Deng et al., 2021). For example, GLIP (Li et al., 2021) reformulates object detection as a phrase grounding task and introduces contrastive training between object regions and language phrases. It shows a great flexibility for heterogeneous datasets and remarkable performance on both closed-set and open-set detection. Despite its impressive results, GLIP's performance can be constrained since it is designed based on a traditional one-stage detector Dynamic Head (Dai et al., 2021a). As open-set and closed-set detection are closely related, we believe a stronger closed-set object detector can result in an even better open-set detector.

Motivated by the encouraging progress of Transformer-based detectors (Zhang et al., 2022a; Liu et al., 2022; Li et al., 2022b; 2023a), in this work, we propose to build a strong open-set detector based on DINO (Zhang et al., 2022a), which not only offers the state-of-the-art object detection performance, but also allows us to integrate multi-level text information into its algorithm by grounded pre-training. We name the model as **Grounding DINO**. Grounding DINO has several advantages over GLIP. First, its Transformer-based architecture is similar to language models, making it easier to process both image and language data. For example, as all the image and language branches are built with Transformers, we can easily fuse cross-modality features in its whole pipeline. Second, Transformer-based detectors have demonstrated a superior capability of leveraging large-scale datasets. Lastly, as a DETR-like model, DINO can be optimized end-to-end without using any hand-crafted modules such as NMS (Non-Maximum Suppression), which greatly simplifies the overall grounding model.

Most existing open-set detectors are developed by extending closed-set detectors to open-set scenarios with language information. As shown in Fig. 2, a closed-set detector typically has three important modules, a backbone for feature extraction, a neck for feature enhancement, and a head for region refinement (or box prediction). A closed-set detector can be generalized to detect novel objects by learning language-aware region embeddings so that each region can

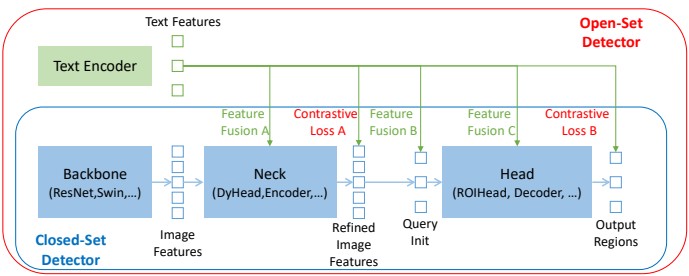

Figure 2: Existing approaches to extending closed-set detectors to open-set scenarios. Note that some closed-set detectors can have only partial phases of the figure.

be classified into novel categories in a language-aware semantic space. The key to achieving this goal is using contrastive loss between region outputs and language features at the neck and/or head outputs. To help a model align cross-modality information, some work tried to fuse features before the final loss stage. Fig. 2 shows that feature fusion can be performed in three phases: neck (phase A), query initialization (phase B), and head (phase C). For example, GLIP (Li et al., 2021) performs early fusion in the neck module (phase A), and OV-DETR (Zang et al., 2022) uses language-aware queries as head inputs (phase B).

We argue that more feature fusion in the pipeline enables the model to perform better. It is worth noting that retrieval tasks prefer a CLIP-like two-tower architecture which only performs multi-modality feature comparison at the end for efficiency. However, for open-set detection, the model is normally given both an image and a text input that specifies the target object categories or a specific object. In such a case, a tight (and early) fusion model is more preferred for a better performance (Anderson et al., 2017; Li et al., 2021) as both image and text are available at beginning. Although conceptually simple, it is hard for previous work to perform feature fusion in all three phases. The design of classical detectors like Faster RCNN makes it hard to interact with language information in most blocks. Unlike classical detectors, the Transformer-based detector DINO has a consistent structure with language blocks. The layer-by-layer design enables it to interact with language

---

[1]We view the terms *open-set object detection*, *open-world object detection*, and *open-vocabulary object detection* the same task in this paper. To avoid confusion, we always use *open-set object detection* in our paper.

information easily. Under this principle, we design three feature fusion approaches in the neck, query initialization, and head phases. More specifically, we design a feature enhancer by stacking self-attention, text-to-image cross-attention, and image-to-text cross-attention as the neck module. We then develop a language-guided query selection method to initialize queries for head. We also design a cross-modality decoder for the head phase with image and text cross-attention layers to boost query representations. The three fusion phases effectively help the model achieve better performance on existing benchmarks, which will be shown in Sec. 4.4.

Although significant improvements have been achieved in multi-modal learning, most existing open-set detection work evaluates their models on objects of novel categories, as shown in the left column of Fig. 1 (b). We argue that another important scenario, where objects are described with attributes, should also be considered. In the literature, the task is named Referring Expression Comprehension (REC) (Miao et al., 2022; Liu et al., 2017)[2]. We present some examples of REC in the right column of Fig. 1 (b). It is a closely related field but tends to be overlooked in previous open-set detection work. In this work, we extend open-set detection to support REC and also evaluate its performance on REC datasets.

We conduct experiments on all three settings, including closed-set detection, open-set detection, and referring object detection, to comprehensively evaluate open-set detection performance. Grounding DINO outperforms competitors by a large margin. For example, Grounding DINO reaches a 52.5 AP on COCO minival without any COCO training data. It also establishes a new state of the art on the ODinW (Li et al., 2022a) zero-shot benchmark with a 26.1 mean AP.

| Model | Model Design | | | Text Prompt | Closed-Set Settings | Zero-Shot Transfer | | | Referring Detection |
| | Base Detector | Fusion (Fig. 2) | CLIP | Represent. Level (Sec. 3.4) | COCO | COCO | LVIS | ODinW | RefCOCO/+/g |
|---|---|---|---|---|---|---|---|---|---|
| ViLD (Gu et al., 2021) | Mask R-CNN | - | ✓ | sentence | ✓ | partial label | partial label | | |
| RegionCLIP (Zhong et al., 2022) | Faster RCNN | - | ✓ | sentence | ✓ | partial label | partial label | | |
| FindIt (Kuo et al., 2022) | Faster RCNN | A | | sentence | ✓ | partial label | | | fine-tune |
| MDETR (Kamath et al., 2021) | DETR | A,C | | word | | | fine-tune | zero-shot | fine-tune |
| DQ-DETR (Shilong et al., 2023) | DETR | A,C | | word | ✓ | | zero-shot | | fine-tune |
| GLIP (Li et al., 2021) | DyHead | A | | word | ✓ | zero-shot | zero-shot | zero-shot | |
| GLIPv2 (Zhang et al., 2022c) | DyHead | A | | word | ✓ | zero-shot | zero-shot | zero-shot | |
| OV-DETR (Zang et al., 2022) | Deformable DETR | B | ✓ | sentence | ✓ | partial label | partial label | | |
| OWL-ViT (Minderer et al., 2022) | - | - | ✓ | sentence | ✓ | partial label | partial label | zero-shot | |
| DetCLIP (Yao et al., 2022) | ATSS | - | ✓ | sentence | | | zero-shot | zero-shot | |
| OmDet (Zhao et al., 2022) | Sparse R-CNN | C | ✓ | sentence | ✓ | | | zero-shot | |
| Grounding DINO (Ours) | DINO | A,B,C | | sub-sentence | ✓ | zero-shot | zero-shot | zero-shot | zero-shot |

Table 1: A comparison of previous open-set object detectors. Our summarization is based on the experiments in their paper, but not the ability to extend their models to other tasks. It is worth noting that some related works may not (only) be designed for the open-set object detection initially, like MDETR (Kamath et al., 2021) and GLIPv2(Zhang et al., 2022c), but we list them here for a comprehensive comparison with existing work. We use the term "partial label" for the settings, where models are trained on partial data (e.g. base categories) and evaluated on other cases. (Zareian et al., 2021)

## 2 RELATED WORK

**Detection Transformers.** Grounding DINO is built upon the DETR-like model DINO (Zhang et al., 2022a), which is an end-to-end Transformer-based detector. DETR was first proposed in (Carion et al., 2020) and then has been improved from many directions (Zhu et al., 2021; Meng et al., 2021; Gao et al., 2021b; Dai et al., 2021a; Wang et al., 2021; Jia et al., 2022; Chen et al., 2022) in the past few years. DAB-DETR (Liu et al., 2022) introduces anchor boxes as DETR queries for more accurate box prediction. DN-DETR (Li et al., 2022b) proposes a query denoising approach to stabilizing the bipartite matching. DINO (Zhang et al., 2022a) further develops several techniques including contrastive de-noising and set a new record on the COCO object detection benchmark. However, such detectors mainly focus on closed-set detection and are difficult to generalize to novel classes because of the limited pre-defined categories.

**Open-Set Object Detection.** Open-set object detection is trained using existing bounding box annotations and aims at detecting arbitrary classes with the help of language generalization. OV-DETR (Zareian et al., 2021) uses image and text embedding encoded by a CLIP model as queries to decode the category-specified boxes in the DETR framework (Carion et al., 2020). ViLD (Gu

---

[2]We use the term *Referring Expression Comprehension (REC)* and *Referring (Object) Detection* exchangeable in this paper.

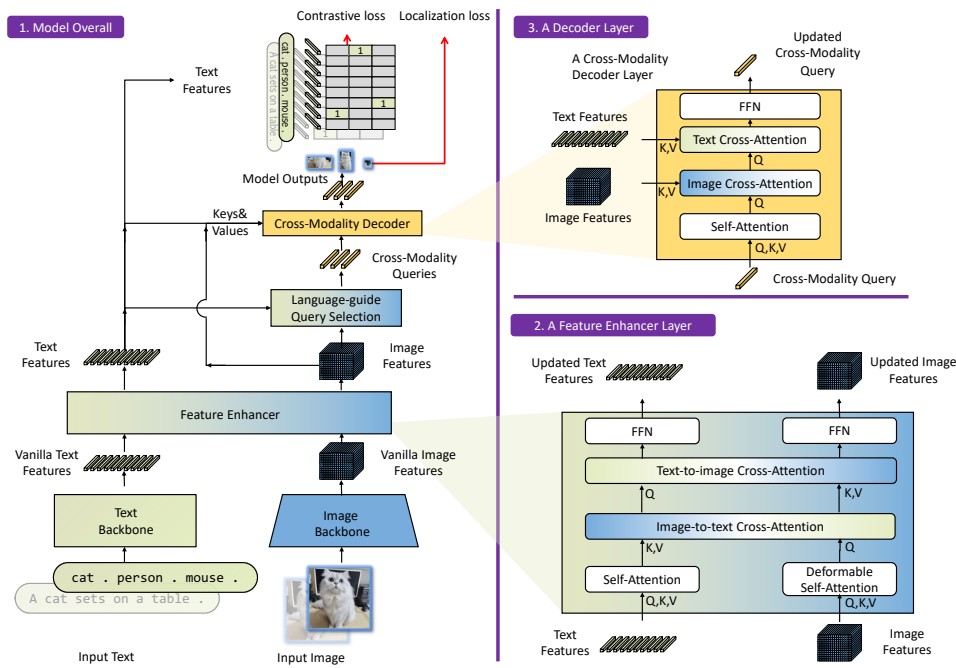

Figure 3: The framework of Grounding DINO. We present the overall framework, a feature enhancer layer, and a decoder layer in block 1, block 2, and block 3, respectively.

et al., 2021) distills knowledge from a CLIP teacher model into a R-CNN-like detector so that the learned region embeddings contain the semantics of language. GLIP (Gao et al., 2021a) formulates object detection as a grounding problem and leverages additional grounding data to help learn aligned semantics at phrase and region levels. It shows that such a formulation can even achieve stronger performance on fully-supervised detection benchmarks. DetCLIP (Yao et al., 2022) involves large-scale image captioning datasets and uses the generated pseudo labels to expand the knowledge database. The generated pseudo labels effectively help extend the generalization ability.

However, previous works only fuse multi-modal information in partial phases, which may lead to sub-optimal language generalization ability. For example, GLIP only considers fusion in the feature enhancement (phase A) and OV-DETR only injects language information at the decoder inputs (phase B). Moreover, the REC task is normally overlooked in evaluation, which is an important scenario for open-set detection. We compare our model with other open-set methods in Table 1.

## 3    GROUNDING DINO

Grounding DINO outputs multiple pairs of object boxes and noun phrases for a given (`Image`, `Text`) pair. For example, as shown in Fig. 3, the model locates a cat and a table from the input image and extracts word `cat` and `table` from the input text as corresponding labels. Both object detection and REC tasks can be aligned with the pipeline. Following GLIP (Li et al., 2021), we concatenate all category names as input texts for object detection tasks. REC requires a bounding box for each text input. We use the output object with the largest scores as the output for the REC task.

Grounding DINO is a dual-encoder-single-decoder architecture. It contains an image backbone for image feature extraction, a text backbone for text feature extraction, a feature enhancer for image and text feature fusion (Sec. 3.1), a language-guided query selection module for query initialization (Sec. 3.2), and a cross-modality decoder for box refinement (Sec. 3.3).

For each (`Image`, `Text`) pair, we first extract vanilla image features and vanilla text features using an image backbone and a text backbone, respectively. The two vanilla features are fed into a feature enhancer module for cross-modality feature fusion. After obtaining cross-modality text and image features, we use a language-guided query selection module to select cross-modality queries from image features. Like the object queries in most DETR-like models, these cross-modality queries

will be fed into a cross-modality decoder to probe desired features from the two modal features and update themselves. The output queries of the last decoder layer will be used to predict object boxes and extract corresponding phrases.

## 3.1 FEATURE EXTRACTION AND ENHANCER

Given an `(Image, Text)` pair, we extract multi-scale image features with an image backbone like Swin Transformer (Liu et al., 2021), and text features with a text backbone like BERT (Devlin et al., 2018). Following previous DETR-like detectors (Zhu et al., 2021; Zhang et al., 2022a), multi-scale features are extracted from the outputs of different blocks. After extracting vanilla image and text features,

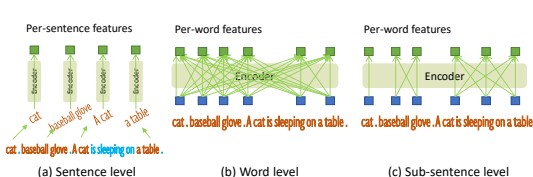

Figure 4: Comparisons of text representations.

we fed them into a feature enhancer for cross-modality feature fusion. The feature enhancer includes multiple feature enhancer layers. We illustrate a feature enhancer layer in Fig. 3 block 2. We leverage the Deformable self-attention to enhance image features and the vanilla self-attention for text feature enhancers. Inspired by GLIP (Li et al., 2021), we add an image-to-text and a text-to-image cross-attention modules for feature fusion. These modules help align features of different modalities.

## 3.2 LANGUAGE-GUIDED QUERY SELECTION

Grounding DINO aims to detect objects from an image specified by an input text. To effectively leverage the input text to guide object detection, we design a language-guided query selection module to select features that are more relevant to the input text as decoder queries.

Let's denote the image feature as $\mathbf{X}_I \in \mathrm{R}^{N_I \times d}$ and the text features as $\mathbf{X}_T \in \mathrm{R}^{N_T \times d}$. Here, $N_I$ represents the number of image tokens, $N_T$ indicates the number of text tokens, and $d$ corresponds to the feature dimension. In our experiments, we specifically utilize a feature dimension of $d = 256$. Typically, in our models, the value of $N_I$ exceeds $10,000$, while $N_T$ remains below $256$. Our objective is to extract $N_q$ queries from the encoder's image features to be used as inputs for the decoder. In alignment with the DINO method, we set $N_q$ to be 900. The top $N_q$ query indices for the image feature, denoted as $\mathbf{I}_{N_q}$, are selected using the following expression:

$$\mathbf{I}_{N_q} = \mathrm{Top}_{N_q}(\mathrm{Max}^{(-1)}(\mathbf{X}_I \mathbf{X}_T^{\intercal})). \tag{1}$$

In this expression, $\mathrm{Top}_{N_q}$ represents the operation to pick the top $N_q$ indices. The function $\mathrm{Max}^{(-1)}$ executes the `max` operation along the $-1$ dimension, and the symbol $\intercal$ denotes matrix transposition. We present the query selection process in Algorithm 1 in PyTorch style. The language-guided query selection module outputs $N_q$ indices. We can extract features based on the selected indices to initialize queries. Following DINO (Zhang et al., 2022a), we use mixed query selection to initialize decoder queries. Each decoder query contains two parts: content part and positional part (Meng et al., 2021), respectively. We formulate the positional part as dynamic anchor boxes (Liu et al., 2022), which are initialized with encoder outputs. The other part, the content queries, are set to be learnable during training.

## 3.3 CROSS-MODALITY DECODER

We develop a cross-modality decoder to combine image and text modality features, as shown in Fig. 3 block 3. Each cross-modality query is fed into a self-attention layer, an image cross-attention layer to combine image features, a text cross-attention layer to combine text features, and an FFN layer in each cross-modality decoder layer. Each decoder layer has an extra text cross-attention layer compared with the DINO decoder layer, as we need to inject text information into queries for better modality alignment.

| Model | Backbone | Pre-Training Data | Zero-Shot 2017val | Fine-Tuning 2017val/test-dev |
|---|---|---|---|---|
| Faster R-CNN | RN50-FPN | - | - | 40.2 / - |
| Faster R-CNN | RN101-FPN | - | - | 42.0 / - |
| DyHead-T (Dai et al., 2021a) | Swin-T | - | - | 49.7 / - |
| DyHead-L (Dai et al., 2021a) | Swin-L | - | - | 58.4 / 58.7 |
| DyHead-L (Dai et al., 2021a) | Swin-L | O365,ImageNet21K | - | 60.3 / 60.6 |
| SoftTeacher (Xu et al., 2021) | Swin-L | O365,SS-COCO | - | 60.7 / 61.3 |
| DINO(Swin-L) (Zhang et al., 2022a) | Swin-L | O365 | - | 62.5 / - |
| DyHead-T†(Dai et al., 2021a) | Swin-T | O365 | 43.6 | 53.3 / - |
| GLIP-T (B) (Li et al., 2021) | Swin-T | O365 | 44.9 | 53.8 / - |
| GLIP-T (C) (Li et al., 2021) | Swin-T | O365,GoldG | 46.7 | 55.1 / - |
| GLIP-L (Li et al., 2021) | Swin-L | FourODs,GoldG,Cap24M | 49.8 | 60.8 / 61.0 |
| DINO(Swin-T)†(Zhang et al., 2022a) | Swin-T | O365 | 46.2 | 56.9 / - |
| Grounding DINO T (Ours) | Swin-T | O365 | 46.7 | 56.9 / - |
| Grounding DINO T (Ours) | Swin-T | O365,GoldG | 48.1 | 57.1 / - |
| Grounding DINO T (Ours) | Swin-T | O365,GoldG,Cap4M | 48.4 | 57.2 / - |
| Grounding DINO L (Ours) | Swin-L | O365,OI(Krasin et al., 2017),GoldG | **52.5** | **62.6 / 62.7 (63.0 / 63.0)**\* |
| Grounding DINO L (Ours) | Swin-L | O365,OI,GoldG,Cap4M,COCO,RefC | **60.7** | **62.6** / - |

Table 2: Zero-shot domain transfer and fine-tuning on COCO. * The results in brackets are trained with $1.5\times$ image sizes, i.e., with a maximum image size of 2000. †The models map a subset of O365 categories to COCO for zero-shot evaluations.

## 3.4 SUB-SENTENCE LEVEL TEXT FEATURE

Two kinds of text prompts are explored in previous works, which we named as sentence level representation and word level representation, as shown in Fig. 4. Sentence level representation (Yao et al., 2022; Minderer et al., 2022) encodes a whole sentence to one feature. If some sentences in phrase grounding data have multiple phrases, it extracts these phrases and discards other words. In this way, it removes the influence between words while losing fine-grained information in sentences. Word level representation (Gao et al., 2021a; Kamath et al., 2021) enables encoding multiple category names with one forward but introduces unnecessary dependencies among categories, especially when the input text is a concatenation of multiple category names in an arbitrary order. As shown in Fig. 4 (b), some unrelated words interact during attention. To avoid unwanted word interactions, we introduce attention masks to block attentions among unrelated category names, named "sub-sentence" level representation. It eliminates the influence between different category names while keeping per-word features for fine-grained understanding.

## 3.5 LOSS FUNCTION

Following previous DETR-like works (Carion et al., 2020; Zhu et al., 2021; Meng et al., 2021; Liu et al., 2022; Li et al., 2022b; Zhang et al., 2022a), we use the L1 loss and the GIOU (Rezatofighi et al., 2019) loss for bounding box regressions. We follow GLIP (Li et al., 2021) and use contrastive loss between predicted objects and language tokens for classification. Specifically, we dot product each query with text features to predict logits for each text token and then compute focal loss (Lin et al., 2017) for each logit. Box regression and classification costs are first used for bipartite matching between predictions and ground truths. We then calculate final losses between ground truths and matched predictions with the same loss components. Following DETR-like models, we add auxiliary loss after each decoder layer and after the encoder outputs.

## 4 EXPERIMENTS

## 4.1 SETUP

We conduct extensive experiments on three settings: a closed-set setting on the COCO detection benchmark (Sec. D.1), an open-set setting on zero-shot COCO, LVIS, and ODinW (Sec. 4.2), and a referring detection setting on RefCOCO/+/g (Sec. 4.3). Ablations are then conducted to show the effectiveness of our model design (Sec. 4.4). We also explore a way to transfer a well-trained DINO to the open-set scenario by training a few plug-in modules in Sec. 4.5. The test of our model efficiency is presented in Sec. J.

**Implementation Details** We trained two model variants, Grounding DINO T with Swin-T (Liu et al., 2021), and Grounding DINO L with Swin-L (Liu et al., 2021) as an image backbone, respectively. We leveraged BERT-base (Devlin et al., 2018) from Hugging Face (Wolf et al., 2019) as text backbones. As we focus more on the model performance on novel classes, we list zero-shot transfer and referring detection results in the main text. More implementation details are available in the Appendix Sec. B.

## 4.2 ZERO-SHOT TRANSFER OF GROUNDING DINO

In this setting, we pre-train models on large-scale datasets and directly evaluate models on new datasets. We also list some fine-tuned results for a more thorough comparison of our model with prior works.

**COCO Benchmark** We compare Grounding DINO with GLIP and DINO in Table 2. We pre-train models on large-scale datasets and directly evaluate our model on the COCO benchmark. As the O365 dataset (Shao et al., 2019) has (nearly[3]) covered all categories in COCO, we evaluate an O365 pre-trined DINO on COCO as a zero-shot baseline. The result shows that DINO performs better on the COCO zero-shot transfer than DyHead. Grounding DINO outperforms all previous models on the zero-shot transfer setting, with $+0.5$AP and $+1.8$AP compared with DINO and GLIP under the same setting. Grounding data is still helpful for Grounding DINO, introducing more than 1AP (48.1 vs. 46.7) on the zero-shot transfer setting. With stronger backbones and larger data, Grounding DINO sets a new record of 52.5 AP on the COCO object detection benchmark without seeing any COCO images during training. Grounding DINO obtains a 62.6 AP on COCO minival, outperforming DINO's 62.5 AP. When enlarging the input images by $1.5\times$, the benefits reduce. We suspect that the text branch enlarges the gap between models with different input images. Even though the performance plateaus with larger input size, Grounding DINO gets an impressive 63.0 AP on COCO test-dev with fine-tuning on the COCO dataset(See the number in brackets of Table 2).

**LVIS Benchmark** LVIS (Gupta et al., 2019) is a dataset for long-tail objects. It contains more than 1000 categories for evaluation. We use LVIS as a downstream task to test the zero-shot abilities of our model. We use GLIP and DetCLIPv2 as baselines for our models. The results are shown in Table 3.

We found two interesting phenomena in the results. First, Grounding DINO works better than common objects than GLIP, but worse on rare categories. The other phenomenon is that Grounding DINO has larger gains with more data than GLIP. For example, Grounding DINO introduces $+1.8$ AP gains with the caption data Cap4M, whereas GLIP has only $+1.1$ AP. We believe that Grounding DINO has better scalability compared with GLIP. A larger-scale training will be left as our future work.

| Model | Backbone | Pre-Training Data | MiniVal (Kamath et al., 2021) AP | APr/APc/APf |
|---|---|---|---|---|
| *Zero-Shot Setting* | | | | |
| GLIP-T (C) | Swin-T | O365,GoldG | 24.9 | 17.7/19.5/31.0 |
| GLIP-T | Swin-T | O365,GoldG,Cap4M | 26.0 | 20.8/21.4/31.0 |
| DetCLIPv2 | Swin-T | O365,GoldG,CC15M | 40.4 | 36.0/41.7/40.0 |
| Grounding DINO T | Swin-T | O365,GoldG | 25.6 | 14.4/19.6/32.2 |
| Grounding DINO T | Swin-T | O365,GoldG,Cap4M | 27.4 | 18.1/23.3/32.7 |
| Grounding DINO L | Swin-L | O365,OI,GoldG,Cap4M, COCO,RefC | 33.9 | 22.2/30.7/38.8 |
| *Fine-Tune Setting* | | | | |
| MDETR | RN101 | GoldG,RefC | 24.2 | 20.9/24.9/24.3 |
| Mask R-CNN | RN101 | - | 33.3 | 26.3/34.0/33.9 |
| DetCLIPv2 (Yao et al., 2023) | Swin-T | O365,GoldG,CC15M | 50.7 | 44.3/52.4/50.3 |
| Grounding DINO T | Swin-T | O365,GoldG | **52.1** | 35.4/51.3/55.7 |

Table 3: Model results on LVIS.

To assess whether the number of queries affects performance, further ablations were conducted on query numbers as detailed in Sec. K. The findings indicate that the impact varies with the training dataset. Specifically, models trained exclusively on O365 experience a decline in performance as the number of queries increases. In contrast, models trained on both O365 and GoldG demonstrate improved performance with an increased number of queries.

Although achieving better results than GLIP, we found that Grounding DINO is inferior to DetCLIPv2, which is trained on a larger scale data. This performance difference might be attributed to the disparity in data distribution between the training dataset and the LVIS dataset.

To unveil the full potential of Grounding DINO, we fine-tuned it on the LVIS dataset. Table 3 highlights the commendable capability of our model. Remarkably, despite being pre-trained only on the O365 and GoldG datasets, Grounding DINO outperforms DetCLIPv2-T by a margin of 1.5 AP.

---

[3]It is not an exact mapping between O365 and COCO categories. We made some approximations during evaluation.

This result shows that Grounding DINO might have learned a better object-level representation which helps yield a better performance after fine-tuning (aligning with the target dataset). In our future work, we will perform more studies, including varying the semantic concept coverage of the training data and increasing the scale of the training data, to further improve the zero-shot generalization performance.

| Model | Language Input | Backbone | Model Size | Pre-Training Data | Test AP$_{average}$ | Test AP$_{median}$ |
|---|---|---|---|---|---|---|
| *Zero-Shot Setting* | | | | | | |
| MDETR (Kamath et al., 2021) | ✓ | ENB5 (Tan & Le, 2019) | 169M | GoldG,RefC | 10.7 | 3.0 |
| OWL-ViT (Minderer et al., 2022) | ✓ | ViT L/14(CLIP) | >1243M | O365, VG | 18.8 | 9.8 |
| GLIP-T (Li et al., 2021) | ✓ | Swin-T | 232M | O365,GoldG,Cap4M | 19.6 | 5.1 |
| OmDet (Zhao et al., 2022) | ✓ | ConvNeXt-B | 230M | COCO,O365,LVIS,PhraseCut | 19.7 | 10.8 |
| GLIPv2-T (Zhang et al., 2022b) | ✓ | Swin-T | 232M | O365,GoldG,Cap4M | 22.3 | 8.9 |
| DetCLIP (Yao et al., 2022) | ✓ | Swin-L | 267M | O365,GoldG,YFCC1M | 24.9 | 18.3 |
| Florence (Yuan et al., 2022) | ✓ | CoSwinH | ≈841M | FLD900M,O365,GoldG | 25.8 | 14.3 |
| Grounding DINO T(Ours) | ✓ | Swin-T | 172M | O365,GoldG | 20.0 | 9.5 |
| Grounding DINO T(Ours) | ✓ | Swin-T | 172M | O365,GoldG,Cap4M | 22.3 | 11.9 |
| Grounding DINO L(Ours) | ✓ | Swin-L | 341M | O365,OI,GoldG,Cap4M,COCO,RefC | **26.1** | **18.4** |
| *Few-Shot Setting* | | | | | | |
| DyHead-T (Dai et al., 2021a) | ✗ | Swin-T | ≈100M | O365 | 37.5 | 36.7 |
| GLIP-T (Li et al., 2021) | ✓ | Swin-T | 232M | O365,GoldG,Cap4M | 38.9 | 33.7 |
| DINO-Swin-T (Zhang et al., 2022a) | ✗ | Swin-T | 49M | O365 | 41.2 | 41.1 |
| OmDet (Zhao et al., 2022) | ✓ | ConvNeXt-B | 230M | COCO,O365,LVIS,PhraseCut | 42.4 | 41.7 |
| Grounding DINO T(Ours) | ✓ | Swin-T | 172M | O365,GoldG | **46.4** | **51.1** |
| *Full-Shot Setting* | | | | | | |
| GLIP-T (Li et al., 2021) | ✓ | Swin-T | 232M | O365,GoldG,Cap4M | 62.6 | 62.1 |
| DyHead-T (Dai et al., 2021a) | ✗ | Swin-T | ≈100M | O365 | 63.2 | 64.9 |
| DINO-Swin-T (Zhang et al., 2022a) | ✗ | Swin-T | 49M | O365 | 66.7 | 68.5 |
| OmDet (Zhao et al., 2022) | ✓ | ConvNeXt-B | 230M | COCO,O365,LVIS,PhraseCut | 67.1 | 71.2 |
| DINO-Swin-L (Zhang et al., 2022a) | ✗ | Swin-L | 218M | O365 | 68.8 | 70.7 |
| Grounding DINO T(Ours) | ✓ | Swin-T | 172M | O365,GoldG | **70.7** | **76.2** |

Table 4: Results on the ODinW benchmark.

**ODinW Benchmark**   ODinW (Object Detection in the Wild) (Li et al., 2022a) is a more challenging benchmark to test model performance under real-world scenarios. It collects more than 35 datasets for evaluation. We report three settings, zero-shot, few-shot, and full-shot results in Table 4. Grounding DINO performs well on this benchmark. With only O365 and GoldG for pre-train, Grounding DINO T outperforms DINO on few-shot and full-shot settings. Impressively, Grounding DINO with a Swin-T backbone outperforms DINO with Swin-L on the full-shot setting.

Grounding DINO outperforms GLIP under the same backbone for the zero-shot setting. Grounding DINO and GLIPv2-T show similar $AP_{average}$. However, a key distinction lies in the $AP_{median}$, where Grounding DINO significantly outperforms GLIPv2-T (11.9 vs 8.9). This suggests that while GLIPv2 may exhibit larger performance variance across different datasets, Grounding DINO maintains a more consistent performance level. GLIPv2 incorporates advanced techniques like masked text training and cross-instance contrastive learning, making it more complex than our Grounding DINO model. Moreover, our model is more compact (172M parameters) compared to GLIPv2 (232M parameters). These factors combined—performance consistency, model complexity, and size—should address concerns about our model's capability in true open-set scenarios.

Grounding DINO L set a new record on ODinW zero-shot with a 26.1 AP, even outperforming the giant Florence models (Yuan et al., 2022). The results show the generalization and scalability of Grounding DINO.

### 4.3   REFERRING OBJECT DETECTION SETTINGS

We further explore our models' performances on the REC task. We leverage GLIP (Li et al., 2021) as our baseline. We evaluate the model performance on RefCOCO/+/g directly.[4] The results are shown in Table 5. Grounding DINO outperforms GLIP under the same setting. Nevertheless, both GLIP and Grounding DINO perform not well without REC data. More training data like caption data or larger models help the final performance, but quite minor. After injecting RefCOCO/+/g data into training, Grounding DINO obtains significant gains. The results reveal that most nowadays open-set object detectors need to pay more attention for a more fine-grained detection.

---

[4]We used the official released code and checkpoints in `https://github.com/microsoft/GLIP`.

| Method | Backbone | Pre-Training Data | Fine-tuning | RefCOCO | | | RefCOCO+ | | | RefCOCOg | |
|---|---|---|---|---|---|---|---|---|---|---|---|
| | | | | val | testA | testB | val | testA | testB | val | test |
| MAttNet (Yu et al., 2018) | R101 | None | ✓ | 76.65 | 81.14 | 69.99 | 65.33 | 71.62 | 56.02 | 66.58 | 67.27 |
| VGTR (Du et al., 2021) | R101 | None | ✓ | 79.20 | 82.32 | 73.78 | 63.91 | 70.09 | 56.51 | 65.73 | 67.23 |
| TransVG (Deng et al., 2021) | R101 | None | ✓ | 81.02 | 82.72 | 78.35 | 64.82 | 70.70 | 56.94 | 68.67 | 67.73 |
| VILLA_L* (Gan et al., 2020) | R101 | CC, SBU, COCO, VG | ✓ | 82.39 | 87.48 | 74.84 | 76.17 | 81.54 | 66.84 | 76.18 | 76.71 |
| RefTR (Li & Sigal, 2021) | R101 | VG | ✓ | 85.65 | 88.73 | 81.16 | 77.55 | 82.26 | 68.99 | 79.25 | 80.01 |
| MDETR (Kamath et al., 2021) | R101 | GoldG,RefC | ✓ | 86.75 | 89.58 | 81.41 | 79.52 | 84.09 | 70.62 | 81.64 | 80.89 |
| DQ-DETR (Shilong et al., 2023) | R101 | GoldG,RefC | ✓ | 88.63 | 91.04 | 83.51 | 81.66 | 86.15 | 73.21 | 82.76 | 83.44 |
| GLIP-T(B) | Swin-T | O365,GoldG | | 49.96 | 54.69 | 43.06 | 49.01 | 53.44 | 43.42 | 65.58 | 66.08 |
| GLIP-T | Swin-T | O365,GoldG,Cap4M | | 50.42 | 54.30 | 43.83 | 49.50 | 52.78 | 44.59 | 66.09 | 66.89 |
| Grounding DINO T (Ours) | Swin-T | O365,GoldG | | 50.41 | 57.24 | 43.21 | 51.40 | 57.59 | 45.81 | 67.46 | 67.13 |
| Grounding DINO T (Ours) | Swin-T | O365,GoldG,RefC | | 73.98 | 74.88 | 59.29 | 66.81 | 69.91 | 56.09 | 71.06 | 72.07 |
| Grounding DINO T (Ours) | Swin-T | O365,GoldG,RefC | ✓ | 89.19 | 91.86 | 85.99 | 81.09 | 87.40 | 74.71 | 84.15 | 84.94 |
| Grounding DINO L (Ours)* | Swin-L | O365,OI,GoldG,Cap4M,COCO,RefC | ✓ | **90.56** | **93.19** | **88.24** | **82.75** | **88.95** | **75.92** | **86.13** | **87.02** |

Table 5: Top-1 accuracy comparison on the referring expression comprehension task. We mark the best results in bold. All models are trained with a ResNet-101 backbone. We use the notations "CC", "SBU", "VG", "OI", "O365", and "YFCC" for Conceptual Captions (Sharma et al., 2018), SBU Captions (Ordonez et al., 2011), Visual Genome (Krishna et al., 2017), OpenImage (Kuznetsova et al., 2018), Objects365 (Zhou et al., 2019), YFCC100M (Thomee et al., 2016) respectively. The term "RefC" is used for RefCOCO, RefCOCO+, and RefCOCOg three datasets. * There might be a data leak since COCO includes validation images in RefC. But the annotations of the two datasets are different.

## 4.4 Ablations

We conduct ablation studies in this section. We propose a tight fusion grounding model for open-set object detection and a sub-sentence level text prompt. To verify the effectiveness of the model design, we remove some fusion blocks for different variants. Results are shown in Table 6. All models are pre-trained on O365 with a Swin-T backbone.

The results show that encoder fusion significantly improves model performance on both COCO and LVIS datasets. The results from comparing model #1 with the baseline model #0 validate this observation. Other techniques, such as language-guided query selection, text cross-attention, and sub-sentence text prompt, also contribute positively to the LVIS performance, yielding significant gains of +3.0 AP, +1.8 AP, and +0.5 AP, respectively. Additionally, these methods enhance the COCO zero-shot performance, further underscoring their effectiveness. However, we observed that language-guided query selection and sub-sentence text prompt had minimal impact on the COCO fine-tune performance. This outcome is reasonable, given that these methods do not alter model parameters or add computational burdens. Text cross-attention, while introducing fewer parameters than encoder fusion, showed less performance improvement compared to encoder fusion (+0.6 vs. +0.8). This finding suggests that fine-tuning performance is predominantly influenced by the model's parameters, indicating that scaling models is a promising direction for enhancing performance.

## 4.5 Transfer from DINO to Grounding DINO

Recent work has presented many large-scale image models for detection with DINO architecture[5]. It is computationally expensive to train a Grounding DINO model from scratch. However, the cost can be significantly reduced if we leverage pre-trained DINO weights. Hence, we conduct some experiments to transfer pre-trained DINO to Grounding DINO models. We freeze the modules co-existing in DINO and Grounding DINO and fine-tune the other parameters only. (We compare DINO and Grounding DINO in Sec. F.) The results are available in Table 7.

| #ID | Model | COCO minival | | LVIS minival |
|---|---|---|---|---|
| | | Zero-Shot | Fine-Tune | Zero-Shot |
| 0 | Grounding DINO (Full Model) | 46.7 | 56.9 | 16.1 |
| 1 | w/o encoder fusion | 45.8 | 56.1 | 13.1 |
| 2 | static query selection | 46.3 | 56.6 | 13.6 |
| 3 | w/o text cross-attention | 46.1 | 56.3 | 14.3 |
| 4 | word-level text prompt | 46.4 | 56.6 | 15.6 |

Table 6: Ablations for our model. All models are trained on the O365 dataset with a Swin Transformer Tiny backbone.

It shows that we can achieve similar performances with Grounding DINO-Training only text and fusion blocks using a pre-trained DINO. Interestingly, the DINO-pre-trained Grounding DINO outperforms standard Grounding DINO on LVIS under the same setting. The results show that there might be much room for model training improvement, which will be our future work to explore.

---

[5]See model instances at `https://github.com/IDEA-Research/detrex`

# 5 CONCLUSION

5!We have presented a Grounding DINO model in this paper. Grounding DINO extends DINO to open-set object detection, enabling it to detect arbitrary objects given texts as queries. We review open-set object detector designs and propose a tight fusion approach to better fusing cross-modality information. We propose a sub-sentence level representation to use detection data for text prompts in a more reasonable way. The results show the effectiveness of our model design and fusion approach. Moreover, we extend open-set object detection to REC tasks and perform evaluation accordingly. We show that existing open-set detectors do not work well for REC data without fine-tuning. Hence we call extra attention to REC zero-shot performance in future studies.

| Model | Pre-Train Data | | COCO minival Zero-Shot | LVIS minival Zero-Shot | ODinW Zero-Shot |
|---|---|---|---|---|---|
| | DINO Pre-Train | Grounded Fine-Tune | | | |
| Grounding DINO T (from scratch) | - | O365 | 46.7 | 16.2 | 14.5 |
| | - | O365,GoldG | 48.1 | 25.6 | 20.0 |
| Grounding DINO T (from pre-trained DINO) | O365 | O365 | 46.5 | 17.9 | 13.6 |
| | O365 | O365,GoldG | 46.4 | 26.1 | 18.5 |

Table 7: Transfer pre-trained DINO to Grounding DINO. We freeze shared modules between DINO and Grounding DINO during grounded fine-tuning. All models are trained with a Swin Transformer Tiny backbone.

**Limitations:** Despite the great performance on open-set object detection setting, Grounding DINO cannot be used for segmentation tasks like GLIPv2. Our training data is less than the largest GLIP model, which may limit our final performance. Moreover, we find that our model will produce false positive results in some cases, which may need more techniques or data to reduce the hallucination.

**Reproducibility** We will make our code open-source, as well as our online demo. Additionally, we will elucidate our model training parameters, implementation details, and details of training data within this paper, as shown in Sec. 4, Sec. B, and Sec. C, to ensure transparency and reproducibility of our model.

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

## A   DINO TRANSFER RESULT

With a pre-trained DINO initialization, the model converges faster than Grounding DINO from scratch, as shown in Fig. 5. Notably, we use the results without exponential moving average (EMA) for the curves in Fig. 5, which results in a different final performance that in Table 7. As the model trained from scratch need more training time, we only show results of early epochs.

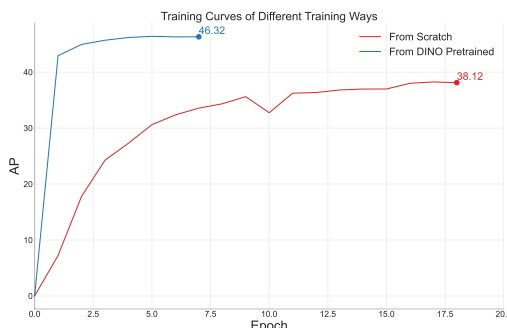

Figure 5: Comparison between two Grounding DINO variants: Training from scratch and transfer from DINO-pretrained models. The models are trained on O365 and evaluated on COCO.

## B MORE IMPLEMENTATION DETAILS

By default, we use 900 queries in our model following DINO. We set the maximum text token number as 256. Using BERT as our text encoder, we follow BERT to tokenize texts with a BPE scheme (Sennrich et al., 2015). We use six feature enhancer layers in the feature enhancer module. The cross-modality decoder is composed of six decoder layers as well. We leverage deformable attention (Zhu et al., 2021) in image cross-attention layers.

Both matching costs and final losses include classification losses (or contrastive losses), box L1 losses, and GIOU (Rezatofighi et al., 2019) losses. Following DINO, we set the weight of classification costs, box L1 costs, and GIOU costs as 2.0, 5.0, and 2.0, respectively, during Hungarian matching. The corresponding loss weights are 1.0, 5.0, and 2.0 in the final loss calculation.

Our Swin Transformer Tiny models are trained on 16 Nvidia V100 GPUs with a total batch size of 32. We extract three image feature scales, from $8\times$ to $32\times$. It is named "4scale" in DINO since we downsample the $32\times$ feature map to $64\times$ as an extra feature scale. For the model with Swin Transformer Large, we extract four image feature scales from backbones, from $4\times$ to $32\times$. The model is trained on 64 Nvidia A100 GPUs with a total batch size of 64.

| Item | Value |
| --- | --- |
| optimizer | AdamW |
| lr | 1e-4 |
| lr of image backbone | 1e-5 |
| lr of text backbone | 1e-5 |
| weight decay | 0.0001 |
| clip max norm | 0.1 |
| number of encoder layers | 6 |
| number of decoder layers | 6 |
| dim feedforward | 2048 |
| hidden dim | 256 |
| dropout | 0.0 |
| nheads | 8 |
| number of queries | 900 |
| set cost class | 1.0 |
| set cost bbox | 5.0 |
| set cost giou | 2.0 |
| ce loss coef | 2.0 |
| bbox loss coef | 5.0 |
| giou loss coef | 2.0 |

Table 8: Hyper-parameters used in our pre-trained models.

### B.1 PSEUDO CODE LANGUAGE-GUIDED QUERY SELECTION

---

**Algorithm 1:** Pseudocode of Language-guided Query Selection in PyTorch-like style.

---

```
"""
Input:
image_feat: (bs, num_img_tokens, ndim)
text_feat: (bs, num_text_tokens, ndim)
num_query: int.

Output:
topk_idx: (bs, num_query)
"""
logits = torch.einsum("bic,btc->bit", image_feat, text_feat) # bs, num_img_tokens,
    num_text_tokens
logits_per_img_feat = logits.max(-1)[0]# bs, num_img_tokens
topk_idx = torch.topk(logits_per_img_feature, num_query, dim=1)[1] # bs, num_query
```

---

The variables `image_feat` and `text_feat` are used for image and text features, respectively. `num_query` is the number of queries in the decoder, which is set to 900 in our implementation. We use `bs` and `ndim` for batch size and feature dimension in the pseudo-code. `num_img_tokens` and `num_text_tokens` are used for the number of image and text tokens, respectively.

## C  DATA USAGE

We use three types of data in our model pre-train.

1. **Detection data.** Following GLIP (Li et al., 2021), we reformulate the object detection task to a phrase grounding task by concatenating the category names into text prompts. We use COCO (Lin et al., 2014), O365 (Shao et al., 2019), and OpenImage(OI) (Krasin et al., 2017) for our model pretrain. To simulate different text inputs, we randomly sampled category names from all categories in a dataset on the fly during training.

2. **Grounding data.** We use the GoldG and RefC data as grounding data. Both GoldG and RefC are preprocessed by MDETR (Kamath et al., 2021). These data can be fed into Grounding DINO directly. GoldG contains images in Flickr30k entities (Plummer et al., 2015a;b) and Visual Genome (Krishna et al., 2017). RefC contains images in RefCOCO, RefCOCO+, and RefCOCOg.

3. **Caption data.** To enhance the model performance on novel categories, we feed the semantic-rich caption data to our model. Following GLIP, we use the pseudo-labeled caption data for model training. In our experiments, we use the same data with GLIP under comparable settings. More specifically, we use GLIP-T annotated caption data for Grounding DINO T, while GLIP-L annotated caption data for Grounding DINO L.

There are two versions of the O365 dataset, which we termed O365v1 and O365v2, respectively. O365v1 is a subset of O365v2. O365v1 contains about 600K images, while O365v2 contains about 1.7M images. Following previous works (Li et al., 2021; Yao et al., 2022), we pre-train the Grounding DINO T on O365v1 for a fair comparison. The Grounding DINO L is pre-trained on O365v2 for a better result.

## D  MORE RESULTS ON COCO DETECTION BENCHMARKS

### D.1  COCO DETECTION RESULTS UNDER THE $1\times$ SETTING

We present the performance of Grounding DINO on standard COCO detection benchmark in Table 9. All models are trained with a ResNet-50 (He et al., 2016) backbone for 12 epochs. Grounding DINO achieves 48.1 AP under the research setting, which shows that Grounding DINO is a strong closed-set detector. However, it is inferior compared with the original DINO. We suspect that the new components may make the model harder to optimize than DINO.

| Model | Epochs | AP | $AP_{50}$ | $AP_{75}$ | $AP_S$ | $AP_M$ | $AP_L$ |
|---|---|---|---|---|---|---|---|
| Faster-RCNN(5scale) (Ren et al., 2017) | 12 | 37.9 | 58.8 | 41.1 | 22.4 | 41.1 | 49.1 |
| DETR(DC5) (Carion et al., 2020) | 12 | 15.5 | 29.4 | 14.5 | 4.3 | 15.1 | 26.7 |
| Deformable DETR(4scale)(Zhu et al., 2021) | 12 | 41.1 | – | – | – | – | – |
| DAB-DETR(DC5)[†] (Liu et al., 2022) | 12 | 38.0 | 60.3 | 39.8 | 19.2 | 40.9 | 55.4 |
| Dynamic DETR(5scale) (Dai et al., 2021b) | 12 | 42.9 | 61.0 | 46.3 | 24.6 | 44.9 | 54.4 |
| Dynamic Head(5scale) (Dai et al., 2021a) | 12 | 43.0 | 60.7 | 46.8 | 24.7 | 46.4 | 53.9 |
| HTC(5scale) (Chen et al., 2019) | 12 | 42.3 | – | – | – | – | – |
| DN-Deformable-DETR(4scale) (Li et al., 2022b) | 12 | 43.4 | 61.9 | 47.2 | 24.8 | 46.8 | 59.4 |
| DINO-4scale (Zhang et al., 2022a) | 12 | 49.0 | 66.6 | 53.5 | 32.0 | 52.3 | 63.0 |
| Grounding DINO (4scale) | 12 | 48.1 | 65.8 | 52.3 | 30.4 | 51.3 | 62.3 |

Table 9: Results for Grounding DINO and other detection models with the ResNet50 backbone on COCO `val2017` trained with 12 epochs (the so called $1\times$ setting).

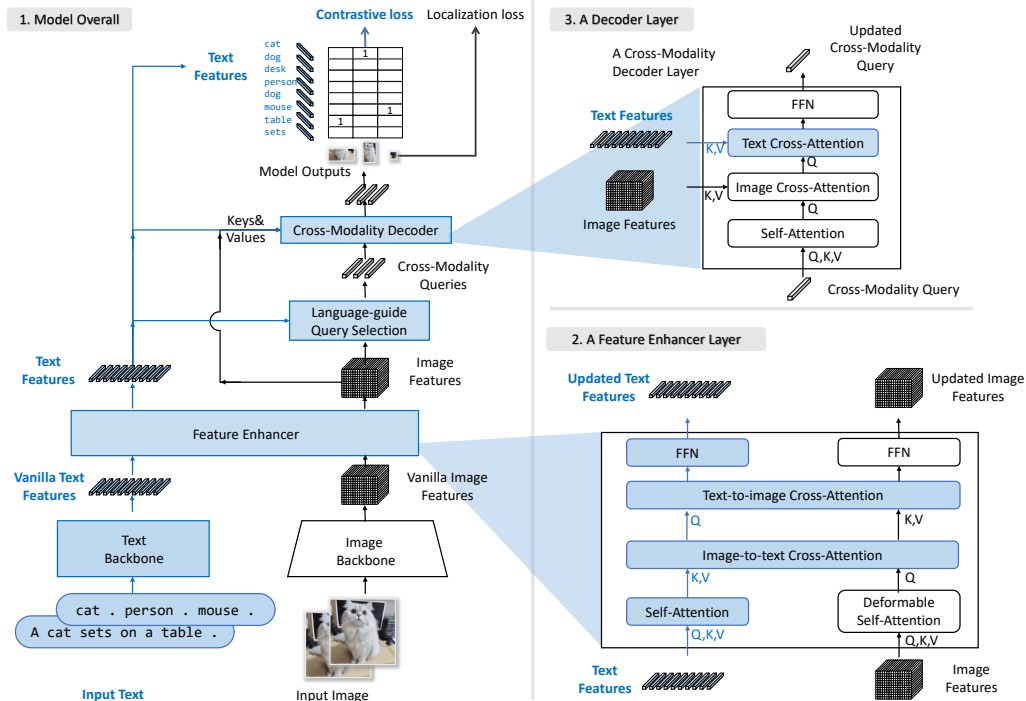

Figure 6: Comparison between DINO and our Grounding DINO. We mark the modifications in blue. Best view in color.

# E    DETAILED RESULTS ON ODINW

We present detailed results of Grounding DINO on ODinW35(Li et al., 2022a) in Table 10, Table 11, and Table 12.

## E.1    COMPARISON BETWEEN GROUNDING DINO AND GLIP ON ODINW

In our comparison of Grounding DINO and GLIP across various datasets in ODinW, as presented in Table 13, we observe that Grounding DINO underperforms on certain uncommon datasets where both models generally show limited effectiveness. For instance, in the PlantDoc dataset, Grounding DINO scores 0.36 compared to GLIP's 1.1. This dataset includes infrequent categories such as "Tomato leaf mosaic virus," which are not well-represented in the training data. These findings highlight the need for improving data quality to enhance overall model performance.

| Dataset | AP | $AP_{50}$ | $AP_{75}$ | $AP_S$ | $AP_M$ | $AP_L$ |
|---|---|---|---|---|---|---|
| AerialMaritimeDrone_large | 9.48 | 15.61 | 8.35 | 8.72 | 10.28 | 2.91 |
| AerialMaritimeDrone_tiled | 17.56 | 26.35 | 13.89 | 0 | 1.61 | 28.7 |
| AmericanSignLanguageLetters | 1.45 | 2.21 | 1.39 | -1 | -1 | 1.81 |
| Aquarium | 18.83 | 34.32 | 18.19 | 10.65 | 20.64 | 21.52 |
| BCCD_BCCD | 6.17 | 11.31 | 6.04 | 1.27 | 9.09 | 6.89 |
| ChessPiece | 6.99 | 11.13 | 9.03 | -1 | -1 | 8.11 |
| CottontailRabbits | 71.93 | 85.05 | 85.05 | -1 | 70 | 73.58 |
| DroneControl_Drone_Control | 6.15 | 10.95 | 6.23 | 2.08 | 6.91 | 6.16 |
| EgoHands_generic | 48.07 | 75.06 | 56.52 | 1.48 | 11.42 | 51.84 |
| EgoHands_specific | 0.66 | 1.25 | 0.64 | 0 | 0.02 | 0.92 |
| HardHatWorkers | 2.39 | 9.17 | 1.07 | 2.13 | 4.32 | 4.6 |
| MaskWearing | 0.58 | 1.43 | 0.56 | 0.12 | 0.51 | 4.66 |
| MountainDewCommercial | 18.22 | 29.73 | 21.33 | 0 | 23.23 | 49.8 |
| NorthAmericaMushrooms | 65.48 | 71.26 | 66.18 | -1 | -1 | 65.49 |
| OxfordPets_by-breed | 0.27 | 0.6 | 0.21 | -1 | 1.38 | 0.33 |
| OxfordPets_by-species | 1.66 | 5.02 | 1 | -1 | 0.65 | 1.89 |
| PKLot_640 | 0.08 | 0.26 | 0.02 | 0.14 | 0.79 | 0.11 |
| Packages | 56.34 | 68.65 | 68.65 | -1 | -1 | 56.34 |
| PascalVOC | 47.21 | 57.59 | 51.28 | 16.53 | 39.51 | 58.5 |
| Raccoon_Raccoon | 44.82 | 76.44 | 46.16 | -1 | 17.08 | 48.56 |
| ShellfishOpenImages | 23.08 | 32.21 | 26.94 | -1 | 18.82 | 23.28 |
| ThermalCheetah | 12.9 | 19.65 | 14.72 | 0 | 8.35 | 50.15 |
| UnoCards | 0.87 | 1.52 | 0.96 | 2.91 | 2.18 | -1 |
| VehiclesOpenImages | 59.24 | 71.88 | 64.69 | 7.42 | 32.38 | 72.21 |
| WildfireSmoke | 25.6 | 43.96 | 25.34 | 5.03 | 18.85 | 42.59 |
| boggleBoards | 0.81 | 2.92 | 0.12 | 2.96 | 1.13 | -1 |
| brackishUnderwater | 1.3 | 1.88 | 1.4 | 0.99 | 1.75 | 11.39 |
| dice_mediumColor | 0.16 | 0.72 | 0.07 | 0.38 | 3.3 | 2.23 |
| openPoetryVision | 0.18 | 0.5 | 0.06 | -1 | 0.25 | 0.17 |
| pistols | 46.4 | 66.47 | 47.98 | 4.51 | 22.94 | 55.03 |
| plantdoc | 0.34 | 0.51 | 0.35 | -1 | 0.28 | 0.86 |
| pothole | 19.87 | 28.94 | 22.23 | 12.49 | 15.6 | 28.78 |
| selfdrivingCa | 9.46 | 19.13 | 8.19 | 0.85 | 6.82 | 16.51 |
| thermalDogsAndPeople | 72.67 | 86.65 | 79.98 | 33.93 | 30.2 | 86.71 |
| websiteScreenshots | 1.51 | 2.8 | 1.42 | 0.85 | 2.06 | 2.59 |

Table 10: Detailed results on 35 datasets in ODinW of Grounding DINO with Swin-T pre-trained on O365 and GoldG.

| Dataset | AP | $AP_{50}$ | $AP_{75}$ | $AP_S$ | $AP_M$ | $AP_L$ |
|---|---|---|---|---|---|---|
| AerialMaritimeDrone_large | 10.3 | 18.17 | 9.21 | 8.92 | 11.2 | 7.35 |
| AerialMaritimeDrone_tiled | 17.5 | 28.04 | 18.58 | 0 | 3.64 | 24.16 |
| AmericanSignLanguageLetters | 0.78 | 1.17 | 0.76 | -1 | -1 | 1.02 |
| Aquarium | 18.64 | 35.27 | 17.29 | 11.33 | 17.8 | 21.34 |
| BCCD_BCCD | 11.96 | 22.77 | 8.65 | 0.16 | 5.02 | 13.15 |
| ChessPiece | 15.62 | 22.02 | 20.19 | -1 | -1 | 15.72 |
| CottontailRabbits | 67.61 | 78.82 | 78.82 | -1 | 70 | 68.09 |
| DroneControl_Drone_Control | 4.99 | 8.76 | 5 | 0.65 | 5.03 | 8.61 |
| EgoHands_generic | 57.64 | 90.18 | 66.78 | 3.74 | 24.67 | 61.33 |
| EgoHands_specific | 0.69 | 1.37 | 0.63 | 0 | 0.02 | 1.03 |
| HardHatWorkers | 4.05 | 13.16 | 1.96 | 2.29 | 7.55 | 9.81 |
| MaskWearing | 0.25 | 0.81 | 0.15 | 0.09 | 0.13 | 2.78 |
| MountainDewCommercial | 25.46 | 39.08 | 28.89 | 0 | 32.53 | 58.38 |
| NorthAmericaMushrooms | 68.18 | 72.89 | 69.75 | -1 | -1 | 68.62 |
| OxfordPets_by-breed | 0.21 | 0.42 | 0.22 | -1 | 2.91 | 0.17 |
| OxfordPets_by-species | 1.3 | 3.95 | 0.71 | -1 | 0.28 | 1.62 |
| PKLot_640 | 0.06 | 0.18 | 0.02 | 0.03 | 0.59 | 0.15 |
| Packages | 60.53 | 76.24 | 76.24 | -1 | -1 | 60.53 |
| PascalVOC | 55.65 | 66.51 | 60.47 | 19.61 | 44.25 | 67.21 |
| Raccoon_Raccoon | 60.07 | 84.81 | 66.5 | -1 | 11.23 | 65.86 |
| ShellfishOpenImages | 29.56 | 38.08 | 33.5 | -1 | 6.38 | 29.95 |
| ThermalCheetah | 17.72 | 25.93 | 19.61 | 1.04 | 20.02 | 63.69 |
| UnoCards | 0.81 | 1.3 | 1 | 2.6 | 1.01 | -1 |
| VehiclesOpenImages | 58.49 | 71.56 | 63.64 | 8.22 | 28.03 | 71.1 |
| WildfireSmoke | 20.04 | 39.74 | 22.49 | 4.13 | 15.71 | 30.41 |
| boggleBoards | 0.29 | 1.15 | 0.04 | 1.8 | 0.57 | -1 |
| brackishUnderwater | 1.47 | 2.34 | 1.58 | 2.32 | 3.31 | 9.96 |
| dice_mediumColor | 0.33 | 1.38 | 0.15 | 0.03 | 1.05 | 12.57 |
| openPoetryVision | 0.05 | 0.19 | 0 | -1 | 0.09 | 0.21 |
| pistols | 66.99 | 86.34 | 72.65 | 16.25 | 39.24 | 75.98 |
| plantdoc | 0.36 | 0.47 | 0.39 | -1 | 0.24 | 0.82 |
| pothole | 25.21 | 38.21 | 26.01 | 8.94 | 18.45 | 39.28 |
| selfdrivingCa | 9.95 | 20.55 | 8.28 | 1.36 | 7.27 | 15.46 |
| thermalDogsAndPeople | 67.89 | 80.85 | 78.66 | 45.05 | 30.24 | 85.56 |
| websiteScreenshots | 1.3 | 2.26 | 1.21 | 0.95 | 1.81 | 2.23 |

Table 11: Detailed results on 35 datasets in ODinW of Grounding DINO with Swin-T pre-trained on O365, GoldG, and Cap4M.

| Dataset | AP | $AP_{50}$ | $AP_{75}$ | $AP_S$ | $AP_M$ | $AP_L$ |
|---|---|---|---|---|---|---|
| AerialMaritimeDrone_large | 12.64 | 18.44 | 14.75 | 9.15 | 19.16 | 0.98 |
| AerialMaritimeDrone_tiled | 20.47 | 34.81 | 12.79 | 0 | 7.61 | 26.93 |
| AmericanSignLanguageLetters | 3.94 | 4.84 | 4 | -1 | -1 | 4.48 |
| Aquarium | 28.14 | 45.47 | 30.97 | 12.1 | 24.71 | 39.42 |
| BCCD_BCCD | 23.85 | 36.92 | 28.88 | 0.3 | 10.8 | 24.43 |
| ChessPiece | 18.44 | 26.3 | 23.33 | -1 | -1 | 18.62 |
| CottontailRabbits | 71.66 | 88.48 | 88.48 | -1 | 66 | 73.04 |
| DroneControl_Drone_Control | 7.16 | 11.56 | 7.67 | 2.29 | 10.6 | 7.68 |
| EgoHands_generic | 52.08 | 81.57 | 59.15 | 1.12 | 31.78 | 55.46 |
| EgoHands_specific | 1.22 | 2.28 | 1.2 | 0 | 0.05 | 1.5 |
| HardHatWorkers | 9.14 | 23.64 | 5.6 | 5.09 | 15.34 | 13.59 |
| MaskWearing | 1.64 | 4.69 | 1.18 | 0.44 | 1.05 | 8.67 |
| MountainDewCommercial | 33.28 | 53.59 | 32.76 | 0 | 35.86 | 80 |
| NorthAmericaMushrooms | 72.33 | 73.18 | 73.18 | -1 | -1 | 72.39 |
| OxfordPets_by-breed | 0.58 | 1.05 | 0.59 | -1 | 4.46 | 0.6 |
| OxfordPets_by-species | 1.64 | 4.8 | 0.87 | -1 | 1.51 | 1.8 |
| PKLot_640 | 0.25 | 0.71 | 0.05 | 0.31 | 1.44 | 0.4 |
| Packages | 63.86 | 76.24 | 76.24 | -1 | -1 | 63.86 |
| PascalVOC | 66.01 | 76.65 | 71.8 | 32.01 | 55.7 | 75.37 |
| Raccoon_Raccoon | 65.81 | 90.39 | 69.93 | -1 | 26 | 68.97 |
| ShellfishOpenImages | 62.47 | 74.25 | 70.07 | -1 | 26 | 63.06 |
| ThermalCheetah | 21.33 | 26.11 | 24.92 | 2.39 | 15.84 | 75.34 |
| UnoCards | 0.52 | 0.84 | 0.66 | 3.02 | 0.92 | -1 |
| VehiclesOpenImages | 62.74 | 75.15 | 67.23 | 10.66 | 47.46 | 76.36 |
| WildfireSmoke | 23.66 | 45.72 | 25.06 | 1.58 | 22.22 | 35.27 |
| boggleBoards | 0.28 | 1.04 | 0.05 | 5.64 | 0.7 | -1 |
| brackishUnderwater | 2.41 | 3.39 | 2.79 | 4.43 | 3.88 | 21.22 |
| dice_mediumColor | 0.26 | 1.15 | 0.03 | 0 | 1.09 | 4.07 |
| openPoetryVision | 0.08 | 0.35 | 0.01 | -1 | 0.15 | 0.11 |
| pistols | 71.4 | 90.69 | 77.21 | 18.74 | 39.58 | 80.78 |
| plantdoc | 2.02 | 2.64 | 2.37 | -1 | 0.5 | 2.82 |
| pothole | 30.4 | 44.22 | 33.84 | 12.27 | 18.84 | 48.57 |
| selfdrivingCa | 9.25 | 17.72 | 8.39 | 1.93 | 7.03 | 13.02 |
| thermalDogsAndPeople | 72.02 | 86.02 | 79.47 | 29.16 | 68.05 | 86.75 |
| websiteScreenshots | 1.32 | 2.64 | 1.16 | 0.79 | 1.8 | 2.46 |

Table 12: Detailed results on 35 datasets in ODinW of Grounding DINO with Swin-L pre-trained on O365, OI, GoldG, Cap4M, COCO, and RefC.

| Metric | GLIP-T | Grounding DINO T |
|---|---|---|
| Average Score (↑) | 19.6 | 22.3 |
| Median Score (↑) | 5.1 | 11.9 |
| AerialMaritimeDrone_large (↑) | 13.70 | 10.30 |
| AerialMaritimeDrone_tiled (↑) | 12.60 | 17.50 |
| AmericanSignLanguageLetters_American_Sign_Language_Letters (↑) | 2.50 | 0.78 |
| Aquarium_Aquarium_Combined (↑) | 18.30 | 18.64 |
| BCCD_BCCD (↑) | 1.00 | 11.96 |
| ChessPieces_Chess_Pieces (↑) | 10.00 | 15.62 |
| CottontailRabbits (↑) | 69.70 | 67.61 |
| DroneControl_Drone_Control (↑) | 5.10 | 4.99 |
| EgoHands_generic (↑) | 50.00 | 57.64 |
| EgoHands_specific (↑) | 0.80 | 0.69 |
| HardHatWorkers (↑) | 3.00 | 4.05 |
| MaskWearing (↑) | 1.10 | 0.25 |
| MountainDewCommercial (↑) | 21.60 | 25.46 |
| NorthAmericaMushrooms_North_American_Mushrooms (↑) | 75.10 | 68.18 |
| OxfordPets_by-breed (↑) | 0.40 | 0.21 |
| OxfordPets_by-species (↑) | 1.10 | 1.30 |
| PKLot_640 (↑) | 0.00 | 0.06 |
| Packages_Raw (↑) | 72.30 | 60.53 |
| PascalVOC (↑) | 56.10 | 55.65 |
| Raccoon_Raccoon (↑) | 57.80 | 60.07 |
| ShellfishOpenImages (↑) | 25.90 | 29.56 |
| ThermalCheetah (↑) | 2.70 | 17.72 |
| UnoCards (↑) | 0.20 | 0.81 |
| VehiclesOpenImages (↑) | 56.00 | 58.49 |
| WildfireSmoke (↑) | 2.30 | 20.04 |
| boggleBoards_416x416AutoOrient_export (↑) | 0.00 | 0.29 |
| brackishUnderwater (↑) | 3.70 | 1.47 |
| dice_mediumColor_export (↑) | 1.10 | 0.33 |
| openPoetryVision (↑) | 0.00 | 0.05 |
| pistols_export (↑) | 49.80 | 66.99 |
| plantdoc (↑) | 1.10 | 0.36 |
| pothole (↑) | 17.20 | 25.21 |
| selfdrivingCar_fixedLarge_export (↑) | 8.00 | 9.95 |
| thermalDogsAndPeople (↑) | 43.70 | 67.89 |
| websiteScreenshots (↑) | 0.50 | 1.30 |

Table 13: Comparison of Grounding DINO and GLIP on ODinW. Both models are trained on O365, GoldG, and Cap4M with Swin-Tiny backbones.

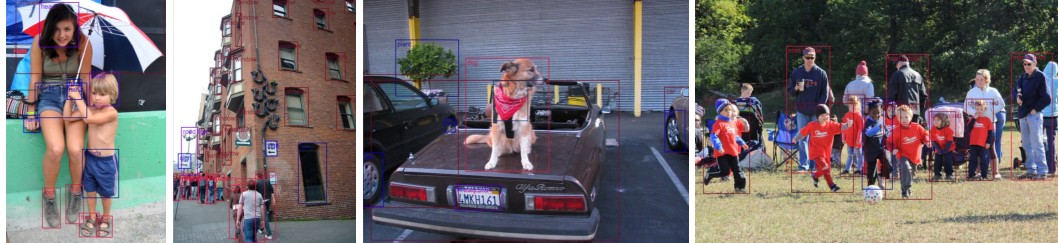

Figure 7: Visualizations of model outputs.

## F COMPARISON BETWEEN DINO AND GROUNDING DINO

To illustrate the difference between DINO and Grounding DINO, we compare DINO and Grounding DINO in Fig. 6. We mark the DINO blocks in gray, while the newly proposed modules are shaded in blue.

| Model | Pre-Train | COCO minival | | LVIS minival | ODinW |
| | | Zero-Shot | Fine-Tune | Zero-Shot | Zero-Shot |
|---|---|---|---|---|---|
| Grounding DINO T | O365,GoldG | 48.1 | 57.1 | 25.6 | 20.0 |
| Grounding DINO T | O365,GoldG,RefC | 48.5 | 57.3 | 21.9 | 17.7 |
| Grounding DINO T | O365,GoldG,RefC,COCO | 56.1 | 57.5 | 22.3 | 17.4 |

Table 14: Impacts of RefC and COCO data for open-set settings. All models are trained with a Swin Transformer Tiny backbone.

## G  VISUALIZATIONS

We present some visualizations in Fig. 7. Our model presents great generalization on different scenes and text inputs. For example, Grounding DINO accurately locates `man in blue` and `child in red` in the last image.

## H  MARRY GROUNDING DINO WITH STABLE DIFFUSION

We present an image editing application in Fig. 1 (b). The results in Fig. 1 (b) are generated by two processes. First, we detect objects with Grounding DINO and generate masks by masking out the detected objects or backgrounds. After that, we feed original images, image masks, and generation prompts to an inpainting model (typical Stable Diffusion (Rombach et al., 2021)) to render new images. We use the released checkpoints in `https://github.com/Stability-AI/stablediffusion` for new image generation. More results are available in Figure 8.

The "detection prompt" is the language input for Grounding DINO, while the "generation prompt" is for the inpainting model.

**Using GLIGEN for Grounded Generation**    To enable fine-grained image editing, we combine the Grounding DINO with GLIGEN (Li et al., 2023b). We use the "phrase prompt" in Figure 9 as the input phrases of each box for GLIGEN.

GLIGEN supports grounding results as inputs and can generate objects on specific positions. We can assign each bounding box an object with GLIGEN, as shown in Figure 9 (c) (d). Moreover, GLIGEN can full fill each bounding box, which results in better visualization, as that in Figure 9 (a) (b). For example, we use the same generative prompt in Figure 8 (b) and Figure 9 (b). The GLIGEN results ensure each bounding box with an object and fulfills the detected regions.

## I  EFFECTS OF REFC AND COCO DATA

We add the RefCOCO/+/g (we note it as "RefC" in tables) and COCO into training in some settings. We explore the influence of these data in Table 14. The results show that RefC helps improve the COCO zero-shot and fine-tuning performance but hurts the LVIS and ODinW results. With COCO introduced, the COCO results is greatly improved. It shows that COCO brings marginal improvements on LVIS and slightly decreases on ODinW.

| Model | params | GFLOPS | FPS |
|---|---|---|---|
| GLIP-T (Li et al., 2021) | 232M | 488G | 6.11 |
| Grounding DINO T (Ours) | 172M | 464G | 8.37 |

Table 15: Comparison of model size and model efficiency between GLIP and Grounding DINO.

## J  MODEL EFFICIENCY

We compare the model size and efficiency between Grounding DINO T and GLIP-T in Table 15. The results show that our model has a smaller parameter size and better efficiency than GLIP.

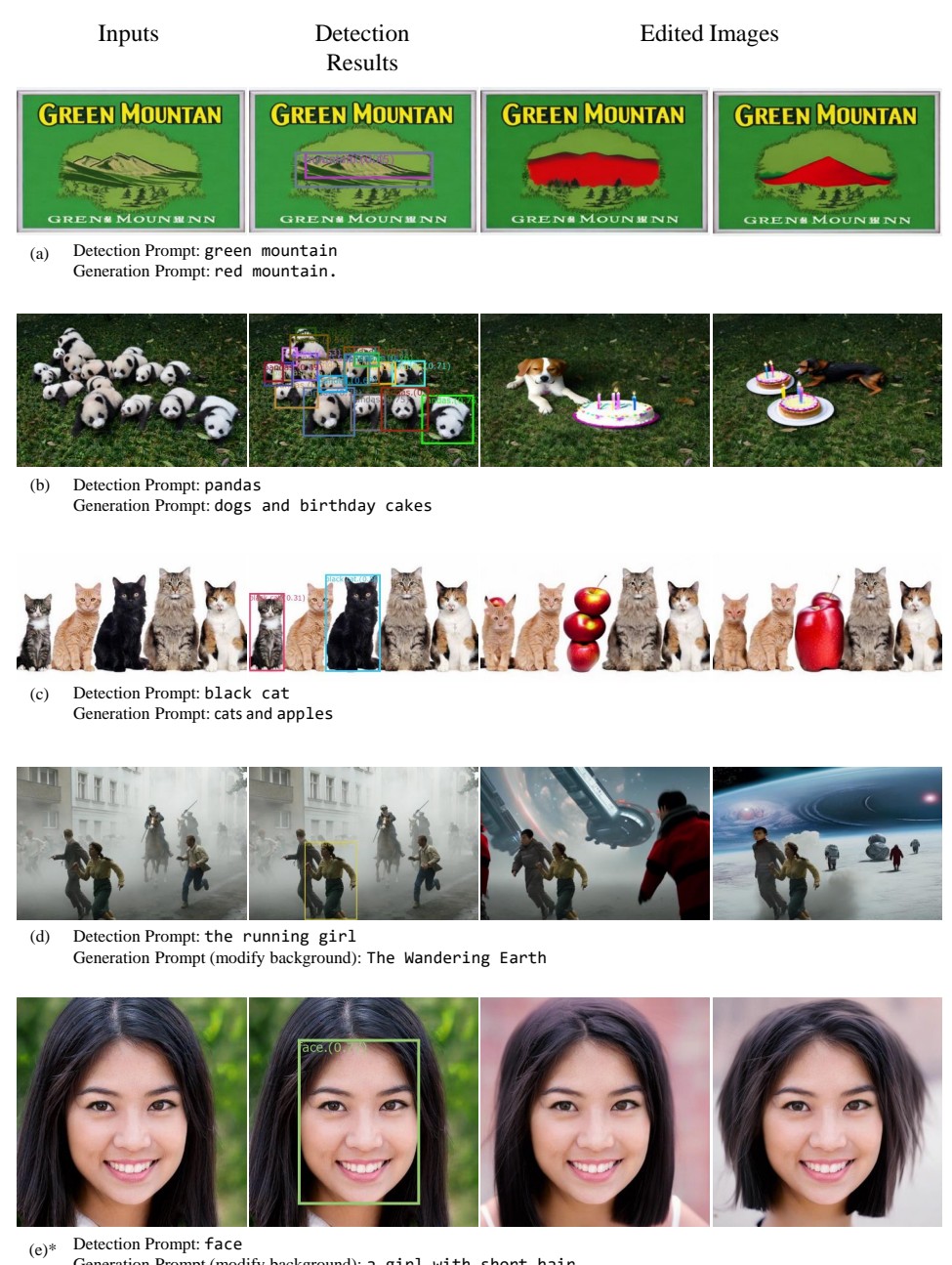

| Inputs | Detection Results | Edited Images |
| --- | --- | --- |

(a) Detection Prompt: green mountain
Generation Prompt: red mountain.

(b) Detection Prompt: pandas
Generation Prompt: dogs and birthday cakes

(c) Detection Prompt: black cat
Generation Prompt: cats and apples

(d) Detection Prompt: the running girl
Generation Prompt (modify background): The Wandering Earth

(e)* Detection Prompt: face
Generation Prompt (modify background): a girl with short hair

Figure 8: Combination of Grounding DINO and Stable Diffusion. We first detect objects with Grounding DINO and then perform image inpainting with Stable Diffusion. "Detection Prompt" and "Generation Prompt" are inputs for Grounding DINO and Stable Diffusion, respectively. *The input human face in the row (e) is generated by StyleGAN.

Inputs    Detection Results    Edited Images

(a) Detection Prompt: `sign`
Generation Prompt: `flying birds.`
Phrase Prompt: `flying birds.`

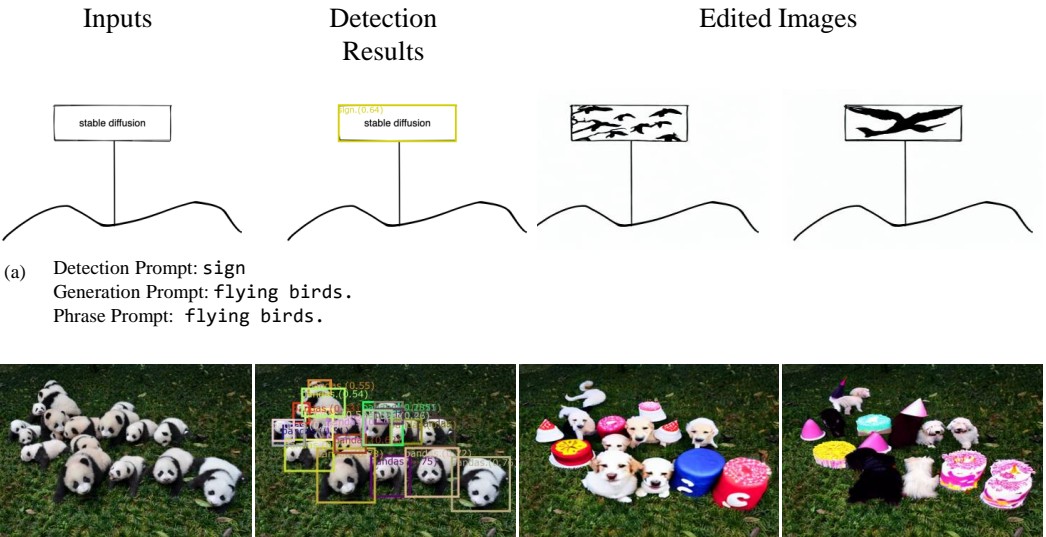

(b) Detection Prompt: `pandas`
Generation Prompt: `dogs and birthday cakes.`
Phrase Prompt*: `a dog; a cake.`

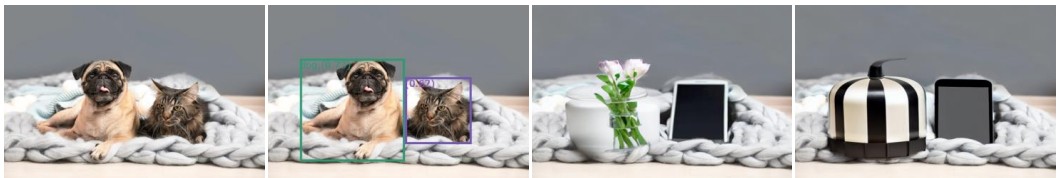

(c) Detection Prompt: `dog, cat`
Generation Prompt: `a cake and a phone`
Phrase Prompt*: `a cake; a phone.`

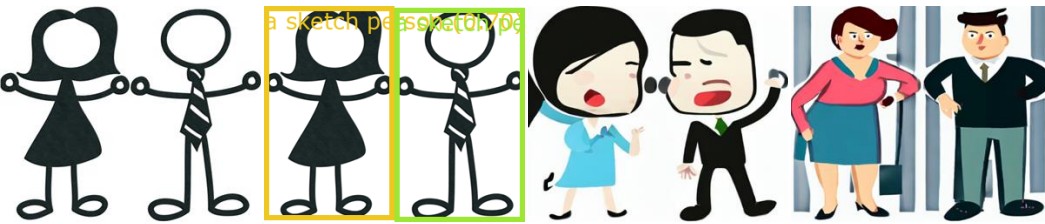

(d) Detection Prompt: `a sketch person`
Generation Prompt: `a woman and a man are talking`
Phrase Prompt*: `a woman; a man.`

Figure 9: Combination of Grounding DINO and GLIGEN. We first detect objects with Grounding DINO and then perform image inpainting with GLIGEN. "Detection Prompt" and "Generation Prompt" are inputs for Grounding DINO and Stable Diffusion, respectively. "Phrase Prompt" are language inputs for each bounding box. The phrase prompts are separated by semicolons. *We assign phrase prompts to bounding boxes randomly.

## K    ABLATIONS FOR MORE DECODER QUERIES

To verify the model performance with more decoder queries, we conducted additional experiments with 1200 and 1500 queries on the COCO and LVIS datasets, detailed in Table R.1 below. We trained two model variants: one on O365 and another on O365+GoldG. Both models were initialized with corresponding checkpoints, except for the learnable queries ('tgt_embed'), to ensure a fair comparison.

For training, the O365 model underwent 3 epochs with a learning rate drop at the end of the 2nd epoch, while the O365+GoldG model was trained for 2 epochs with a learning rate reduction at the end of the 1st epoch. All experiments were carried out on 8xA100 GPUs. It's important to note that due to time and resource constraints, these results might not represent the models' optimal performance. For instance, the LVIS model could potentially benefit from additional training time post-learning rate drop.

The results indicate that models with 1200 and 1500 queries slightly outperform the 900-query version on LVIS rare classes, suggesting better coverage of rare classes. However, the improvement is marginal, as the 900-query model already sufficiently covers all objects in both COCO and LVIS. Additionally, introducing more queries exacerbates data imbalance during training, as the model is trained on objects from sampled categories. This imbalance could offset the benefits of additional queries.

| Pretrain Data | Query Num | COCO | | | | LVIS | | | |
|---|---|---|---|---|---|---|---|---|---|
| | | AP | $AP_S$ | $AP_M$ | $AP_L$ | AP | $AP_r$ | $AP_c$ | $AP_f$ |
| O365+GoldG | 900 | 48.2 | 34.2 | 51.2 | 62.1 | 21.8 | 10.4 | 16.2 | 28.7 |
| O365+GoldG | 1200 | 48.0 | 34.6 | 51.2 | 62.2 | 21.5 | 10.9 | 15.8 | 28.4 |
| O365+GoldG | 1500 | 48.1 | 34.7 | 51.2 | 62.3 | 21.8 | 11.0 | 16.2 | 28.7 |
| O365 | 900 | 46.4 | 33.1 | 49.8 | 60.2 | 14.4 | 6.6 | 8.0 | 21.4 |
| O365 | 1200 | 46.5 | 32.6 | 49.5 | 60.4 | 14.6 | 6.4 | 8.4 | 21.7 |
| O365 | 1500 | 46.3 | 32.7 | 49.3 | 60.3 | 14.8 | 6.3 | 8.6 | 21.8 |

Table 16: Results for Grounding DINO Tiny with more decoder queries

## L    RESULTS WITH DIFFERENT LANGUAGE ENCODER FOR REC

To verify the impacts of language encoders with different sizes, we conducted experiments using two variants of BERT: bert-base-uncased (BERT-B) and bert-large-uncased (BERT-L). These models were trained on a combined dataset consisting of RefCOCO, RefCOCO+, and RefCOCOg. We removed the leaked data in the combined dataset for a fair comparison. It's important to note that training on this combined dataset, as opposed to tuning each dataset separately, might result in slightly lower performance. For a fair comparison, we initialized the models with the O365+GoldG+Cap4M checkpoint, except for the BERT parameters. Limited by time and resources, the training duration was capped at 18 epochs.

Interestingly, our results showed that Grounding DINO with BERT-B outperformed or matched the BERT-L variant in most metrics (as shown in Table 18). This suggests that our default use of BERT-B during the pretrain stage may have contributed to its better performance. Moreover, we didn't observe significant improvements in the late stages of training, indicating that both models were nearing their optimal performance.

This outcome suggests that the main limitation in enhancing REC performance lies within the detection branch, rather than the language processing module. A dedicated model for REC data might be helpful for REC tasks.

## M    MODEL COMPARISONS WITH RELATED WORK

| Model | RefCOCO | | | RefCOCO+ | | | RefCOCOg | |
|---|---|---|---|---|---|---|---|---|
| | val | testA | testB | val | testA | testB | val | test |
| Grounding DINO T (BERT-B) | 87.4 | 91.6 | 84.2 | 78.6 | 86.5 | 73.4 | 81.6 | 83.3 |
| Grounding DINO T (BERT-L) | 87.0 | 91.4 | 84.0 | 78.1 | 86.4 | 73.0 | 81.7 | 83.3 |

Table 17: Results for Grounding DINO with different language encoders

We discuss the similarities and difference of the feature enhancer and cross-modality decoder in Grounding DINO with two recent works: GLIP and X-Decoder (Zou* et al., 2022). In summary, our feature enhancer is similar with GLIP but more reasonable in our pure Transformer architectures. X-Decoder has no similar modules like our feature enhancer. Our cross-modality decoder, especially the text cross-attention, is unique from the designs in GLIP and X-Decoder. Below are detailed comparisons:

**Comparison with GLIP:**

1. *Feature Enhancer:* Our feature enhancer, though similar to GLIP's, is more aligned with our pure Transformer architecture. While GLIP uses DyHead for enhancing visual features, our method employs a Deformable Transformer for image encoder features, ensuring a more consistent architecture across different modalities.

2. *Cross-Modality Decoder:* Unlike GLIP, which uses the same head as ATSS/RetinaNet, our DETR-like model uniquely incorporates a cross-modality decoder. This decoder leverages the Transformer's capability to attend to features from both text and image modalities, a distinction we've validated through our ablation studies.

**Comparison with X-Decoder:**

1. *Feature Fusion:* X-Decoder utilizes a standard image encoder and object decoder for different tasks, without integrating visual and text features during the encoding phase. In contrast, our model employs a feature enhancer for fusing features from both modalities.

2. *Queries in Decoder:* Our model's queries in the decoder aggregate features from both image and text, unlike X-Decoder's approach where the interactions are limited to queries and image features only.

## N   DETIC PSEUDO-LABELED DATA FOR LVIS

To illustrate the potential of our model under similar conditions, we conducted oracle experiments. We pseudo-labeled the ImageNet dataset using a pre-trained Detic(Zhou et al., 2022) model and filtered out LVIS-related images for training, creating a new pseudo-labeled dataset named IN22K-LVIS-1M. It contains about 1M pseudo-labeled images for training. Note that our model under the setting may be not a real zero-shot setting, as our pseudo labeler Detic is trained with LVIS data. The results from these experiments suggest that Grounding DINO can achieve promising LVIS performance, even without direct LVIS training data. A similarly distributed dataset can help the model to generalize well.

| Model | Pre-train Data | LVIS MiniVal - AP(r/c/f) | |
|---|---|---|---|
| | | Zero-shot | Fine-tune |
| DetCLIPv2-T | OG + CC15M | 40.4 (36.0 / 41.7 / 40.0) | 50.7 (44.3 / 52.4 / 50.3) |
| Grounding DINO T | OG+IN22K-LVIS-1M | 40.6 (38.5 / 41.1 / 40.4) | 54.5 (47.3 / 53.9 / 56.1) |

Table 18: Oracle experiments on LVIS. Note that our model under the setting may be not a real zero-shot setting, as our pseudo labeler Detic is trained with LVIS data.

