# OpenReview forum: "Grounding DINO: Marrying DINO with Grounded Pre-Training for Open-Set Object Detection"
_ICLR.cc/2024/Conference — Submitted to ICLR 2024_

### Official Review · Reviewer_9tNN · 2023-10-30

**Soundness:** 2 fair
**Presentation:** 3 good
**Contribution:** 2 fair
**Rating:** 5
**Confidence:** 4

**Summary:**

In this paper, the authors propose an open-vocabulary object detection network namely Grounding DINO. Based on the original DINO object detector, the authors introduce 1) text encoder and cross-modal feature enhancer to incorporate language-specific feature into image features; and 2) language-guided query selection with cross-modal decoder to detect and recognize objects with language guidance. The proposed Grounding DINO obtains promising results on various zero-shot object detection tasks and referring object detection tasks.

**Strengths:**

1. This overall writing is polished and easy to understand.

2. The proposed Grounding DINO shows good generalization ability on various tasks, including zero-shot general object detection and referring object detection.

**Weaknesses:**

1. The key components of proposed method (e.g., feature enhancer and cross-modality decoder) are not new. The authors should thoroughly discuss the relation and difference of between feature enhancer in GLIP [1] and the counterpart in this work, and conduct the same analysis for cross-modality (with X-Decoder [2]).

2. Though utilizing better detector and better pretrained models, this work does not lead to better performance than previous work, e.g., DetCLIP-V2 [3] (CVPR 23), which obtains much better performance on LVIS benchmark and ODinW benchmark than Grounding DINO.

[1] Grounded Language-Image Pre-training, CVPR 2022
[2] Generalized Decoding for Pixel, Image and Language, CVPR 2023
[3] DetCLIPv2: Scalable Open-Vocabulary Object Detection Pre-training via Word-Region Alignment, CVPR 2023

**Questions:**

See weakness section.

---

> ### Author Response · Authors · 2023-11-18
> **Responses to the Reviewer 9tNN (part 1)**
>
> Thanks for your valuable questions. We have added new experiments and discussions in our revised paper. Here are detailed responses.
>
> **1. The key components of the proposed method (e.g., feature enhancer and cross-modality decoder) are not new. The authors should thoroughly discuss the relation and difference of between feature enhancer in GLIP [1] and the counterpart in this work, and conduct the same analysis for cross-modality (with X-Decoder).**
>
>
> Thank you for your insightful questions. In our revised paper, we have included a detailed discussion in Sec. M comparing our method with existing approaches, specifically GLIP and X-Decoder.
>
> In summary, our feature enhancer is similar with GLIP but more reasonable in our pure Transformer architectures. X-Decoder has no similar modules like our feature enhancer. Our cross-modality decoder, especially the text cross-attention, is unique from the designs in GLIP and X-Decoder.
>
> Below are detailed comparisons:
>
> - **Comparison with GLIP:**
>     - **Feature Enhancer:** Our feature enhancer, though similar to GLIP's, is more aligned with our pure Transformer architecture. While GLIP uses DyHead for enhancing visual features, our method employs a Deformable Transformer for image encoder features, ensuring a more consistent architecture across different modalities.
>     - **Cross-Modality Decoder:** Unlike GLIP, which uses the same head as ATSS/RetinaNet, our DETR-like model uniquely incorporates a cross-modality decoder. This decoder leverages the Transformer's capability to attend to features from both text and image modalities, a distinction we've validated through our ablation studies.
>
> - **Comparison with X-Decoder:**
>     - **Feature Fusion:** X-Decoder utilizes a standard image encoder and object decoder for different tasks, without integrating visual and text features during the encoding phase. In contrast, our model employs a feature enhancer for fusing features from both modalities.
>     - **Queries in Decoder:** Our model's queries in the decoder aggregate features from both image and text, unlike X-Decoder's approach where the interactions are limited to queries and image features only.
>
> These comparisons highlight the unique aspects of our method in the context of existing models. We have cited both GLIP and X-Decoder in our revised paper and dedicated a new paragraph to further elaborate on these comparisons.
>
> Thank you again for your valuable questions, which have enriched our paper's discussion section.

---

> ### Author Response · Authors · 2023-11-18
> **Responses to the Reviewer 9tNN (part 2)**
>
> **2. Though utilizing better detector and better pretrained models, this work does not lead to better performance than previous work, e.g., DetCLIP-V2 [3] (CVPR 23), which obtains much better performance on LVIS benchmark and ODinW benchmark than Grounding DINO.**
>
> Thanks for the valuable questions.
>
> **ODinW Benchmark Performance**
>
> It's important to note that **DetCLIP-v2 did not report zero-shot performance on the ODinW benchmark** in their paper, and we could not find their results on the ODinW leaderboard. Without access to their models, code, and training data, it's not feasible for us to independently evaluate their zero-shot performance. However, in terms of fine-tuning performance, **our Grounding DINO Tiny model outperforms DetCLIP-v2 Tiny (Grounding DINO Tiny: 70.0 vs. DetCLIP-v2 Tiny: 68.0) with much less data (Grounding DINO Tiny 5.2M vs. DetCLIP-v2 Tiny: 16.2M data)**. For this comparison, we used the ODinW-13 metric, a subset of the standard ODinW-35 metric, to align with DetCLIP-v2's benchmarking approach.
>
> **LVIS Benchmark Performance**
>
> While DetCLIP-v2 demonstrates better zero-shot performance on the LVIS benchmark, they utilize a substantially larger dataset (\~16.2M data) compared to Grounding DINO (\~5.2M data). This difference in data volume is a significant factor in performance.
>
> To illustrate the potential of our model under similar conditions, we conducted oracle experiments. We pseudo-labeled the ImageNet dataset using a pre-trained Detic model and filtered out LVIS-related images for training, creating a new pseudo-labeled dataset named IN22K-LVIS-1M. It contains about 1M pseudo-labeled images for training. Note that our model under the setting may be not a real zero-shot setting, as our pseudo labeler Detic is trained with LVIS data. The results from these experiments suggest that Grounding DINO can achieve promising LVIS performance, even without direct LVIS training data. A similarly distributed dataset can help the model to generalize well.
>
> Thank you again for raising these points, which have significantly contributed to our paper's thoroughness. We have incorporated these new experiments and findings into our revised paper at Sec. N.
>
> | Model            | Pre-train Data      | LVIS Zeroshot AP(r/c/f)      | LVIS Finetune AP(r/c/f)        |
> |------------------|---------------------|-----------------------------|-------------------------------|
> | DetCLIPv2-T      | OG + CC15M          | 40.4 (36.0 / 41.7 / 40.0)   | 50.7 (44.3 / 52.4 / 50.3)     |
> | Grounding DINO T | OG+IN22K-LVIS-1M    | 40.6 (38.5 / 41.1 / 40.4)   | 54.5 (47.3 / 53.9 / 56.1)     |
>
> Table R.3. Oracle experiments on LVIS.

---

> ### Author Response · Authors · 2023-11-23
> **Please let us know if our responses address your concerns.**
>
> Dear reviewer,
>
> We sincerely appreciate the demands on your time, especially during this busy period. After carefully considering your valuable feedback, We have made some new experiments and necessary modifications to our paper. We are wondering if our responses and revision have addressed your concerns.
>
> We are extremely grateful for your time and effort in reviewing our paper, and we sincerely appreciate your feedback. As today is the last day of the discussion stage, we are kindly awaiting your response.
>
> If you have any further questions or require any additional information from us, please let us know. We would like to invite you to consider our responses and look forward to your reply.
>
> Once again, thank you for your attention and support.
>
> Best regards,
> The Authors

---

### Official Review · Reviewer_sRML · 2023-10-31

**Soundness:** 3 good
**Presentation:** 4 excellent
**Contribution:** 3 good
**Rating:** 8
**Confidence:** 4

**Summary:**

This study proposes a strong open-set object detector called Grounding DINO. When compared with other open-set detectors like GLIP, Grounding DINO introduces the DINO object detector as its overall architecture. It adopts the Swin transformer as the backbone and the encoder-decoder pipeline to replace the DyHead structure as used in GLIP. More fusions between the textual and visual features in the pipeline are also introduced to enable the model to perform better. Moreover, the Referring Expression Comprehension (REC) tasks are also introduced in its pertaining.

Grounding DINO is evaluated on different tasks, including MS-COCO, LVIS, ODinW, and RefCOCO/+/g. It achieves 52.5 AP zero-shot detection on COCO and also obtains state-of-the-art performance on ODinW with 26.1 AP. Albeit strong in common tasks, Grounding DINO performs worse than GLIP on rare classes on OdinW, and the authors also admit its limitation on its zero-shot ability on REC data without fine-tuning.

**Strengths:**

This is a thorough study composed of different shining components.
- First of all, the integration of the DINO object detector pushes forward the performance on different detection benchmarks thanks to the strong feature extraction ability of the new architecture.
- The generalization of different feature fusion methods is clear and inspiring. The proposed feature enhancer calls for more attention to more important features according to the text input. The language-guided query selection module resembles the top-K operation in Efficient DETR/DINO but is endowed with new physical meanings in this pipeline, as only queries that are closely related to the input text are kept.
- The introduction of REC tasks in both pertaining and evaluation opens a new gate for open-set object detectors. The definition of traditional open-sent object detection naturally extends to the fields of describing open-set objects using more versatile expressions.
- The performance reported in this study is competitive. It achieves state-of-the-art performance on the zero-shot ability on MS-COCO and ODinW datasets.
- The reviewer appreciates the presentation regarding the limitations of this study.

**Weaknesses:**

- Novelty issues. Despite the effectiveness of the proposed architectures, the composed modules are not innovative enough as their own. For example, the major component that makes a difference should be the DINO architecture that has already set a record on various detection tasks. The query selection is also similar to that of the top-K operators in Efficient DETR/DINO. Although Grounding DINO extends its tasks to referring expression comprehension, it is only comparable to its precedent studies like GLIP and does not introduce specific modules to address the shortcomings on REC. Nevertheless, I would also admit that these novelty issues are not critical as each of them is not trivial in the task.
- I have some concerns regarding its significance and effectiveness in true open-set scenarios.
Despite the effectiveness of the zero-shot performance on MS-COCO and ODinW datasets, the performance seems to rely on its pertaining data heavily. As also revealed by the author, Grounding DINO transfers better on common classes, classes that are more likely included in the pertaining data in some format, yet performs worse on rare classes in LVIS. The zero-shot performance on the RefCOCO dataset is significantly lower than that after including RefCOCO in its pertaining data.
- A closer look into Table 4 would also reveal that the performance on ODinW is actually on par with GLIPv2 if given the same pertaining data. This would also raise a concern about its true ability in true open-set scenarios.
- Some minor writing issues. e.g., "hard-crafted" -> "hand-crafted" (page 2), "Even though" -> "Even though the performance plateaus with larger input size" (page 7).

**Questions:**

Overall, this is a nice work and an interesting study. I encourage the authors to respond to the above weaknesses.  Besides, I also have some additional questions regarding the paper and possibly some future work.
- It is assumed that the query selection module might be the major reason for the low performance on rare classes in LVIS. Have the authors increased the top-K values to validate the assumption?
- As for the limited ability of the REC dataset, is it possible to increase the model size of the BERT text encoder to obtain a better representation of complex text input to alleviate this?

---

> ### Author Response · Authors · 2023-11-18
> **Responses to the Reviewer sRML (part 1)**
>
> Thank you for your insightful and encouraging feedback. We have incorporated additional experiments on REC and revised the paper based on your writing suggestions. Here is our detailed response:
>
> **1.Novelty issues.** Despite the effectiveness of the proposed architectures, the composed modules are not innovative enough as their own. For example, the major component that makes a difference should be the DINO architecture that has already set a record on various detection tasks. The query selection is also similar to that of the top-K operators in Efficient DETR/DINO. Although Grounding DINO extends its tasks to referring expression comprehension, it is only comparable to its precedent studies like GLIP and does not introduce specific modules to address the shortcomings on REC. Nevertheless, I would also admit that these novelty issues are not critical as each of them is not trivial in the task.
>
> **Response:** We appreciate your acknowledgment of the effectiveness of our proposed architecture. While it is true that the major components, such as the DINO architecture and query selection, resemble existing methods like those in Efficient DETR/DINO, our work uniquely integrates these elements. Specifically, we are the first to adapt DETR-like models to grounded training for open-set detection tasks, thereby achieving superior performance over traditional convolution-based open-set detectors like GLIP.
>
> Our approach may not introduce specific modules for addressing shortcomings in Referring Expression Comprehension (REC), but its strength lies in the effective amalgamation of existing designs. This synthesis itself, we believe, is a meaningful contribution to the field, demonstrating that a well-orchestrated combination of established techniques can yield significant advancements in performance.
>
>
> **2. Effectiveness in true open-set scenarios.** I have some concerns regarding its significance and effectiveness in true open-set scenarios. Despite the effectiveness of the zero-shot performance on MS-COCO and ODinW datasets, the performance seems to rely on its pertaining data heavily. As also revealed by the author, Grounding DINO transfers better on common classes, classes that are more likely included in the pertaining data in some format, yet performs worse on rare classes in LVIS. The zero-shot performance on the RefCOCO dataset is significantly lower than that after including RefCOCO in its pertaining data.
>
> **Response:** Thank you for your insightful and professional comments regarding our model's effectiveness in true open-set scenarios. We acknowledge that while our model demonstrates strong zero-shot performance on the MS-COCO and ODinW datasets, its reliance on common object classes is evident. These common objects, frequently used in various scenarios, underscore the practical utility of our model in many real-world applications.
>
> Regarding the performance on ODinW, which encompasses 35 datasets across diverse domains, our model shows commendable versatility. Some of these datasets feature uncommon objects, further demonstrating the model's capability in more varied open-set environments.
>
> Moreover, we recognize the challenges in evaluating open-set models, a complexity mirrored in fields like image generation and Large Language Models (LLMs). A more comprehensive evaluation set might indeed be necessary for a thorough comparison of open-set models.
>
>
> We do agree that Grounding DINO is less effective in Referring Expression Comprehension (REC) tasks, primarily because it wasn't specifically designed for REC.  However, it shows promise in REC tasks when fine-tuned or pre-trained with REC data, highlighting the critical role of data diversity in open-set models. To address this, we plan to continue enriching our model with a broader range of data to cover more diverse scenarios.
>
> Additionally, our model demonstrates strong few-shot learning capabilities in the ODinW benchmark, significantly outperforming previous works. This efficiency in adapting to new scenarios with minimal data is a key advantage, offering lower costs and greater flexibility compared to earlier models.
>
> In summary, while our model may not be perfect in every open-set scenario, it stands as a robust option for real-world applications, especially given its excellent few-shot performance and adaptability to diverse situations.

---

> ### Author Response · Authors · 2023-11-18
> **Responses to the Reviewer sRML (part 2)**
>
> **3. Compared with GLIPv2 on ODinW.** A closer look into Table 4 would also reveal that the performance on ODinW is actually on par with GLIPv2 if given the same pertaining data. This would also raise a concern about its true ability in true open-set scenarios.
>
> **Response:** Thank you for your valuable questions regarding the performance comparison in Table 4.
>
> Regarding the comparison with GLIPv2, it's true that Grounding DINO and GLIPv2-T show similar $AP_{average}$. However, a key distinction lies in the $AP_{median}$, where Grounding DINO significantly outperforms GLIPv2-T (11.9 vs 8.9). This suggests that while GLIPv2 may exhibit larger performance variance across different datasets, Grounding DINO maintains a more consistent performance level. This consistency is particularly relevant in true open-set scenarios, where variance can greatly impact overall effectiveness.
>
> Additionally, it's important to note the complexity and size differences between the models. GLIPv2 incorporates advanced techniques like masked text training and cross-instance contrastive learning, making it more complex than our Grounding DINO model. Moreover, our model is more compact (172M parameters) compared to GLIPv2 (232M parameters). These factors combined—performance consistency, model complexity, and size—should address concerns about our model's capability in true open-set scenarios.
>
> We have included this analysis in the discussion paragraph of Sec. 4.3 of our revised paper. Thanks again for your questions, which help to make our paper clear.
>
> **4.Some minor writing issues.** e.g., "hard-crafted" -> "hand-crafted" (page 2), "Even though" -> "Even though the performance plateaus with larger input size" (page 7).
>
> **Response:** Thanks for your careful reading and valuable feedbacks! We have revised these typos in our updated version.
>
> **5.It is assumed that the query selection module might be the major reason for the low performance on rare classes in LVIS. Have the authors increased the top-K values to validate the assumption?**
>
> Thank you for the insightful question. In response, we conducted additional experiments with 1200 and 1500 queries on the COCO and LVIS datasets, detailed in Table R.2.1 below. We trained two model variants: one on O365 and another on O365+GoldG. Both models were initialized with corresponding checkpoints, except for the learnable queries (`tgt_embed`), to ensure a fair comparison.
>
>
> For training, the O365 model underwent 3 epochs with a learning rate drop at the end of the 2nd epoch, while the O365+GoldG model was trained for 2 epochs with a learning rate reduction at the end of the 1st epoch. All experiments were carried out on 8xA100 GPUs. It's important to note that due to time and resource constraints, these results might not represent the models' optimal performance. For instance, the LVIS model could potentially benefit from additional training time post-learning rate drop.
>
> From our analysis, we observe that models with 1200 and 1500 queries show a modest improvement in performance on LVIS's rare classes compared to the 900-query model. This suggests an improved coverage of rare classes. However, this improvement is marginal and the overall performances of models with varying query numbers are quite similar. This is likely because the 900-query model already provides ample coverage of LVIS objects. Furthermore, increasing the number of queries introduces a training challenge due to exacerbated data imbalances, as models are trained on objects from sampled categories. This imbalance could negate the benefits of additional queries.
>
> We appreciate your valuable suggestions and have included these results and discussions in the Sec. K of the revised version of our paper.
>
>
> | Model | Query Num | COCO AP | AP-small | AP-medium | AP-large | LVIS AP | AP-r | AP-c | AP-f |
> |:----:|:----:|:----:|:----:|:----:|:----:|:----:|:----:|:----:|:----:|
> | O365+GoldG | 900 | 48.2 | 34.2 | 51.2 | 62.1 | 21.8 | 10.4 | 16.2 | 28.7 |
> | O365+GoldG  | 1200 | 48.0 | 34.6 | 51.2 | 62.2 | 21.5 | 10.9 | 15.8 | 28.4 |
> | O365+GoldG | 1500 | 48.1 | 34.7 | 51.2 | 62.3 | 21.8 | 11.0 | 16.2 | 28.7 |
> | O365 | 900 | 46.4 | 33.1 | 49.8 | 60.2 | 14.4 | 6.6 | 8.0 | 21.4 |
> | O365 | 1200 | 46.5 | 32.6 | 49.5 | 60.4 | 14.6 | 6.4 | 8.4 | 21.7 |
> | O365 | 1500 | 46.3 | 32.7 | 49.3 | 60.3 | 14.8 | 6.3 | 8.6 | 21.8 |
>
> Table R.2.1 Results of the extended experiments with increased query numbers.

---

> ### Author Response · Authors · 2023-11-18
> **Responses to the Reviewer sRML (part 3)**
>
> **6. As for the limited ability of the REC dataset, is it possible to increase the model size of the BERT text encoder to obtain a better representation of complex text input to alleviate this?**
>
> Thank you for your insightful question about enhancing the REC dataset's performance through a larger BERT text encoder model.
>
> We conducted experiments using two variants of BERT: bert-base-uncased (BERT-B) and bert-large-uncased (BERT-L). These models were trained on a combined dataset consisting of RefCOCO, RefCOCO+, and RefCOCOg. We removed the leaked data in the combined dataset for a fair comparison. It's important to note that training on this combined dataset, as opposed to tuning each dataset separately, might result in slightly lower performance. For a fair comparison, we initialized the models with the O365+GoldG+Cap4M checkpoint, except for the BERT parameters. Limited by time and resources, the training duration was capped at 18 epochs.
>
>
> Interestingly, our results showed that Grounding DINO with BERT-B outperformed or matched the BERT-L variant in most metrics (as shown in Table R.2.2). This suggests that our default use of BERT-B during the pre-train stage may have contributed to its better performance. Moreover, we didn't observe significant improvements in the late stages of training, indicating that both models were nearing their optimal performance.
>
>
> This outcome suggests that the main limitation in enhancing REC performance lies within the detection branch, rather than the language processing module. A dedicated model for REC data might be helpful for REC tasks.
>
> We have added the new results to the revised version of our paper at Sec. L. Thanks again for your insightful questions, which have significantly enriched our analysis.
>
>
> | Model | refococo (val) | refococo (testA) | refococo (testB) | refococo+ (val) | refococo+ (testA) | refococo+ (testB) | refococog (val) | refococog (test) |
> |-------|----------------|------------------|------------------|-----------------|-------------------|-------------------|-----------------|------------------|
> | Grounding DINO T (BERT-B) | 87.4 | 91.6 | 84.2 | 78.6 | 86.5 | 73.4 | 81.6 | 83.3 |
> | Grounding DINO T (BERT-L) | 87.0 | 91.4 | 84.0 | 78.1 | 86.4 | 73.0 | 81.7 | 83.3 |

---

> ### Author Response · Authors · 2023-11-23
> **Please let us know if our responses address your concerns.**
>
> Dear reviewer,
>
> We sincerely appreciate the demands on your time, especially during this busy period. After carefully considering your valuable feedback, We have made some new experiments and necessary modifications to our paper. We are wondering if our responses and revision have addressed your concerns.
>
> We are extremely grateful for your time and effort in reviewing our paper, and we sincerely appreciate your feedback. As today is the last day of the discussion stage, we are kindly awaiting your response.
>
> If you have any further questions or require any additional information from us, please let us know. We would like to invite you to consider our responses and look forward to your reply.
>
> Once again, thank you for your attention and support.
>
> Best regards,
> The Authors

---

> > ### Comment · Reviewer_sRML · 2023-11-23
> > **Thank the authors for providing more experimental results**
> >
> > I would like to thank the authors for providing additional experiments to clarify the questions raised in the initial review. Most of my concerns are well addressed. Based on the new results, I would like to encourage the authors to present a deeper analysis to explain the phenomenon observed in the experiments before rushing to make an assumption (e.g. reasons for the performance drop for rare classes).

---

> > > ### Author Response · Authors · 2023-11-23
> > > **Thanks for the valuable suggestions and reminds!**
> > >
> > > Thank you for your valuable feedback, which has greatly assisted in enhancing the comprehensiveness of our paper. We apologize for the issue in the initial version and have accordingly revised the section on LVIS experiments (Sec. 4.2).
> > >
> > > With superior overall zero-shot performance relative to GLIP, Grounding DINO emerges as a preferable option for real-world applications, since average performance is crucial. Moreover, all models show limited effectiveness in entirely new scenarios, which means the gains may not be useful in the real world as well. Luckily, Grounding DINO's stronger function-fitting ability renders it more effective in few-shot settings. These properties make the Grounding DINO a good model to use in real-world scenarios.
> > >
> > > When fine-tuning the model on LVIS, we found Grounding DINO outperforms both GLIPv2 and DetCLIPv2 with a large margin (Grounding DINO 52.1 vs GLIPv2 50.6 & DetCLIPv2 50.7). It works better than GLIPv2 on the rare categories with more training data (Grounding DINO 47.3 vs DetCLIPv2 44.3), as shown in Table 18. This underscores Grounding DINO's potential, suggesting that its lower zero-shot score on LVIS rare classes may be attributable to data limitations.
> > >
> > > Grounding DINO is characterized as a robust function fitter, excelling in generalizing across common and frequent categories, albeit at the expense of reduced out-of-domain capabilities. This reflects a broader trend, where transformer-based detectors, like Grounding DINO, require more data than convolution-based models like GLIP. A similar pattern is noted in the classification domain, with models like ViT compared to CNNs. Increasing query numbers improves the model's generalization to familiar objects but may diminish its performance on rarer classes as well, as shown in the experiments of Table 16.
> > >
> > > To corroborate this, we introduced a new section (E.1) comparing Grounding DINO and GLIP across various ODinW datasets. Here, Grounding DINO tends to underperform on exceptionally uncommon datasets, where both models struggle, such as in the PlantDoc dataset (Grounding DINO 0.36 vs. GLIP 1.1). These datasets often contain rare categories like "Tomato leaf mosaic virus," underscoring the necessity of improving data quality for enhanced model performance.
> > >
> > > We appreciate your suggestions, which have substantially contributed to the improvement of our paper. Due to the limited time of the discussion period, we cannot add more content now. We promise to add more visualizations and analysis on LVIS rare categories to make the paper more comprehensive. The quality of our work is our top priority, and we are immensely grateful for your valuable feedback.

---

### Official Review · Reviewer_jXst · 2023-11-01

**Soundness:** 2 fair
**Presentation:** 2 fair
**Contribution:** 1 poor
**Rating:** 5
**Confidence:** 3

**Summary:**

This paper introduces Grounding DINO, an open-set object detector that combines DINO with grounded pre-training. This allows it to detect objects based on human inputs like category names or expressions.
The innovation lies in using language to enhance closed-set detectors for better open-set detection. The detection process is conceptually divided into three phases, including a feature enhancer, language-guided query selection, and a cross-modality decoder for merging vision and language.
This work also evaluates referring expression comprehension for attribute-specified objects. Grounding DINO excels in multiple benchmarks, including COCO and ODinW, setting new records like a 52.5 AP on COCO's zero-shot transfer benchmark without using its training data.

**Strengths:**

1. The paper shows that text and vision fusion and multiple stages helps the model achieve better performance compared to later fusion.
2. The paper shows benefits on multiple settings, including closed-set detection, open-set detection, and referring object detection, to comprehensively evaluate open-set detection performance
3. Grounding DINO outperforms competitors by a large margin and establishes a new state of the art on the ODinW (zero-shot benchmark with a 26.1 mean AP

**Weaknesses:**

1. More than 900 queries would be interesting to see if the model generalizes well to rare classes as well
2. The ablations in table 6 and 7 do not show strong contribution of individual modeling choices except tight fusion. The authors dont explain this

**Questions:**

Please refer to weakness 1 and 2

---

> ### Author Response · Authors · 2023-11-18
> **Response to the Reviewer jXst**
>
> We thank all your valuable comments of our paper. We have revised the paper and added more experiments as you suggested. The detailed responses are shown below.
>
>
>
> **Q1: More than 900 queries would be interesting to see if the model generalizes well to rare classes as well**
>
> A: Your question about the influence of increasing query numbers on model generalization, especially regarding rare classes, is indeed pivotal. To address this, we expanded our experiments to include 1200 and 1500 queries on both the COCO and LVIS datasets. The results of these experiments are presented in Table R.1.
>
> We trained two model variants, O365 and O365+GoldG, each initialized from respective checkpoints. To maintain consistency, we kept the learnable queries (tgt_embed) unchanged. The training protocols were slightly varied: the O365 model underwent three epochs with a learning rate reduction after the second epoch, while the O365+GoldG model had two epochs with a similar rate adjustment after the first epoch. All tests were conducted using 8xA100 GPUs. We must note that time and resource constraints may have limited our ability to fully optimize the models, particularly for the LVIS dataset. Extended training periods post-learning rate adjustment could potentially enhance performance further.
>
>
> From our analysis, we observe that models with 1200 and 1500 queries show a modest improvement in performance on LVIS's rare classes compared to the 900-query model. This suggests an improved coverage of rare classes. However, this improvement is marginal and the overall performances of models with varying query numbers are quite similar. This is likely because the 900-query model already provides ample coverage of LVIS objects. Furthermore, increasing the number of queries introduces a training challenge due to exacerbated data imbalances, as models are trained on objects from sampled categories. This imbalance could negate the benefits of additional queries.
>
> We appreciate your valuable suggestions and have included these results and discussions in the Sec. K of the revised version of our paper.
>
>
> | Model | Query Num | COCO AP | AP-small | AP-medium | AP-large | LVIS AP | AP-r | AP-c | AP-f |
> |:----:|:----:|:----:|:----:|:----:|:----:|:----:|:----:|:----:|:----:|
> | O365+GoldG | 900 | 48.2 | 34.2 | 51.2 | 62.1 | 21.8 | 10.4 | 16.2 | 28.7 |
> | O365+GoldG  | 1200 | 48.0 | 34.6 | 51.2 | 62.2 | 21.5 | 10.9 | 15.8 | 28.4 |
> | O365+GoldG | 1500 | 48.1 | 34.7 | 51.2 | 62.3 | 21.8 | 11.0 | 16.2 | 28.7 |
> | O365 | 900 | 46.4 | 33.1 | 49.8 | 60.2 | 14.4 | 6.6 | 8.0 | 21.4 |
> | O365 | 1200 | 46.5 | 32.6 | 49.5 | 60.4 | 14.6 | 6.4 | 8.4 | 21.7 |
> | O365 | 1500 | 46.3 | 32.7 | 49.3 | 60.3 | 14.8 | 6.3 | 8.6 | 21.8 |
>
> Table R.1 Results of the extended experiments with increased query numbers.
>
>
> **Q2: The ablations in table 6 and 7 do not show strong contribution of individual modeling choices except tight fusion.**
>
> A: Thank you for raising these important points regarding the ablations in Tables 6 and 7.
>
> Regarding Table 7, our primary aim was to demonstrate an efficient approach for Grounding DINO training. As detailed in Section 4.5, we illustrate that freezing a pre-trained DINO and tuning additional parameters introduced in Grounding DINO can achieve faster model convergence and comparable performance to full-parameter pretrained checkpoints.
>
> Addressing your concerns about Table 6, our data shows that encoder fusion (tight fusion) significantly improves model performance on both COCO and LVIS datasets. The results from comparing model #1 with the baseline model #0 validate this observation. Other techniques, such as language-guided query selection, text cross-attention, and sub-sentence text prompt, also contribute positively to the LVIS performance, yielding significant gains of +3.0 AP, +1.8 AP, and +0.5 AP, respectively. Additionally, these methods enhance the COCO zero-shot performance, further underscoring their effectiveness.
>
> However, we observed that language-guided query selection and sub-sentence text prompt had minimal impact on the COCO fine-tune performance. This outcome is reasonable, given that these methods do not alter model parameters or add computational burdens. Text cross-attention, while introducing fewer parameters than encoder fusion, showed less performance improvement compared to encoder fusion (+0.6 vs. +0.8). This finding suggests that fine-tuning performance is predominantly influenced by the model's parameters, indicating that scaling models is a promising direction for enhancing performance.
>
> We have included a detailed discussion of these points in Section 4.4 of our revised paper. Thank you again for your valuable questions, which have helped to improve our paper.

---

> ### Author Response · Authors · 2023-11-23
> **Please let us know if our responses address your concerns.**
>
> Dear reviewer,
>
> We sincerely appreciate the demands on your time, especially during this busy period. After carefully considering your valuable feedback, We have made some new experiments and necessary modifications to our paper. We are wondering if our responses and revision have addressed your concerns.
>
> We are extremely grateful for your time and effort in reviewing our paper, and we sincerely appreciate your feedback. As today is the last day of the discussion stage, we are kindly awaiting your response.
>
> If you have any further questions or require any additional information from us, please let us know. We would like to invite you to consider our responses and look forward to your reply.
>
> Once again, thank you for your attention and support.
>
> Best regards,
> The Authors

---

### Author Response · Authors · 2023-11-18
**Paper Revision**

We express our sincere gratitude to all reviewers for their constructive and insightful comments. Your suggestions have been invaluable in enhancing the quality of our paper. We have carefully revised the manuscript, incorporating your feedback. For ease of review, all updates are marked in purple (violet).

**Detailed Updates:**

1. **Increased Query Experiments**: Following suggestions from Reviewers jXst and sRML, we have added experiments with more than 900 queries in Section K.
2. **Ablation Discussions**: Thanks to Reviewer jXst's suggestion, we've included detailed discussions about the ablations.
3. **Performance Analysis**: Thanks to Reviewer sRML's suggestion, Section 4.3 now features an in-depth performance discussion of Grounding DINO and GLIPv2.
4. **Correction of Typos**: We appreciate Reviewer sRML pointing out writing errors, which we have diligently corrected.
5. **Large BERT Model Experiments**: Pursuant to Reviewer sRML's recommendation, experiments with large BERT models are now detailed in Section L.
6. **Comparative Analysis**: In Section M, we've added a comprehensive comparison with GLIP and X-Decoder, focusing on feature enhancer and cross-modality decoder. This enhancement comes thanks to Reviewer 9tNN.
7. **Additional LVIS Data Experiments**: Following Reviewer 9tNN's advice, we conducted experiments with more pseudo-labeled data on LVIS, detailed in Section N.
8: **Updated LVIS Experiments Analysis**: Following Reviewer sRML's advice, we modified the analysis of more queries in Sec 4.2.
9: **Detailed Result Comparison with GLIP**: Following Reviewer sRML's advice, we added a comparison with GLIP on ODinW in Sec E.1.

Thanks again for all the reviewers' valuable suggestions, which helped us to further improve the paper. Please feel free to share your further concerns and suggestions, and we will continue to improve the paper.

---

### Meta-Review · Area_Chair_cQ1Y · 2023-12-10

**Metareview:**

Paper proposes an open-set object detector, termed Grounding DINO. The design is a combination of Transformer-based detector DINO with grounded pre-training. The key to this design is an effective fusion of language and vision modalities. The design is composed of three components: (a) a feature enhancer that cross-attends between language and visual features, (b) a language-guided query selection, and (c) cross-modal decoder. Paper presents strong performance on open-set object detection and referring expression comprehension.

Three reviewers reviewed the submitted paper, resulting in ranking of: 2 x Marginally below the acceptance threshold and 1 x Accept, good paper. Reviewers agree that the paper is well written and accessible and presented idea is sound. The main concerns were focused on (1) delineation of claims and design with respect to GLIP and X-Decoder, (2) discussion of ablation that appears to show that tight fusion is what mostly accounts for improved performance, and (3) novelty (which was echoed by [sRML] (which was the most positive reviewer) and [9tNN]). Authors provided an effective rebuttal that addressed most of the concerns with experimental design.

AC has read the reviews, rebuttal and discussion that followed as well as most of the paper itself. This appears to be a very Borderline case, where while the paper is well written and approach is sound, design is well motivated and results are compelling, the novelty as mentioned by  [sRML] and [9tNN] and also, to a large extent, acknowledged by authors themselves ("strength lies in the effective amalgamation of existing designs") is limited. While agglomeration of design is a fine contribution when it bring about new insights, it is hard to draw such insights from the proposed design in the paper. The fact that cross-attention between language and vision is powerful is already known in the community. So the main insight appears that a particular combination of well chosen components results in state-of-the-art performance. While this is good, it is at the same time, a bit underwhelming.

**Justification For Why Not Higher Score:**

To be honest, this is a Borderline paper and I can easily go with Accept (poster). I do think the novelty is a bit limited and insights are thin, but the paper is well written and executed. The performance is also very competitive.

**Justification For Why Not Lower Score:**

N/A

---

### Decision · Program_Chairs · 2024-01-16

Reject